# Dichotomous role of the human mitochondrial Na⁺/Ca2⁺/Li⁺ exchanger NCLX in colorectal cancer growth and metastasis

Trayambak Pathak[1†], Maxime Gueguinou[1†], Vonn Walter[2,3,4], Celine Delierneux[1], Martin T Johnson[1], Xuexin Zhang[1], Ping Xin[1], Ryan E Yoast[1], Scott M Emrich[1], Gregory S Yochum[3,5], Israel Sekler[6], Walter A Koltun[5], Donald L Gill[1], Nadine Hempel[1,4,7], Mohamed Trebak[1]*

[1]Department of Cellular and Molecular Physiology, The Pennsylvania State University College of Medicine, Hershey, United States; [2]Department of Public Health Sciences, The Pennsylvania State University College of Medicine, Hershey, United States; [3]Department of Biochemistry and Molecular Biology, The Pennsylvania State University College of Medicine, Hershey, United States; [4]Penn State Cancer Institute. The Pennsylvania State University College of Medicine, Hershey, United States; [5]Department of Surgery, Division of Colon and Rectal Surgery, The Pennsylvania State University College of Medicine, Hershey, United States; [6]Department of Physiology and Cell Biology, Faculty of Health Sciences, Ben-Gurion University of the Negev, Beer Sheva, Israel; [7]Department of Pharmacology, The Pennsylvania State University College of Medicine, Hershey, United States

*For correspondence:
mtrebak@psu.edu

†These authors contributed equally to this work

Competing interests: The authors declare that no competing interests exist.

**Abstract** Despite the established role of mitochondria in cancer, the mechanisms by which mitochondrial $Ca^{2+}$ ($mtCa^{2+}$) regulates tumorigenesis remain incompletely understood. The crucial role of $mtCa^{2+}$ in tumorigenesis is highlighted by altered expression of proteins mediating $mtCa^{2+}$ uptake and extrusion in cancer. Here, we demonstrate decreased expression of the mitochondrial $Na^+/Ca^{2+}/Li^+$ exchanger NCLX (*SLC8B1*) in human colorectal tumors and its association with advanced-stage disease in patients. Downregulation of NCLX causes $mtCa^{2+}$ overload, mitochondrial depolarization, decreased expression of cell-cycle genes and reduced tumor size in xenograft and spontaneous colorectal cancer mouse models. Concomitantly, NCLX downregulation drives metastatic spread, chemoresistance, and expression of epithelial-to-mesenchymal, hypoxia, and stem cell pathways. Mechanistically, $mtCa^{2+}$ overload leads to increased mitochondrial reactive oxygen species, which activate HIF1$\alpha$ signaling supporting metastasis of NCLX-null tumor cells. Thus, loss of NCLX is a novel driver of metastasis, indicating that regulation of $mtCa^{2+}$ is a novel therapeutic approach in metastatic colorectal cancer.

## Introduction

Mitochondria are adaptable cellular organelles critical for a spectrum of essential functions including ATP generation, cell signaling, metabolism, proliferation, and death (*Vyas et al., 2016*). This central function causes mitochondria to fulfill a crucial role as mediators of tumorigenesis. Mitochondria sense changes in energetic, biosynthesis, and cellular stress, and adapt to the surrounding tumor environment to modulate cancer progression and drug resistance (*Tosatto et al., 2016*; *Vyas et al.,*

**eLife digest** Colorectal cancer is the second largest cause of cancer deaths worldwide. Even in cases where the cancer is diagnosed and treated early, cells can sometimes survive treatment and spread to other organs. Once the cancer has spread, the survival rate is less than 15%.

Mitochondria are compartments in the cell that produce energy, and they play an important role in supporting the rapid growth of cancer cells. The levels of calcium ions in mitochondria control how they produce energy, a process that is altered in cancer cells. To better understand how calcium ions influence colorectal cancer growth, Pathak, Gueguinou et al. studied a protein called NCLX, which controls calcium levels by pumping them out of the mitochondria.

Two mouse strains that were used to study what happens if NCLX is missing. The first strain was genetically modified to disable the gene for NCLX and then exposed to carcinogens. The second strain was injected with colorectal cancer cells from a human tumor that were lacking NCLX. In both strains, the tumors that formed were smaller than in mice with NCLX. However, the human cancer cells in the second model were more likely to spread to other organs. This is likely because the build-up of calcium ions in the mitochondria of mice lacking NCLX led to an increase in the production of hypoxia-inducible factor-1a, a protein that is a common driver of cancer spread.

Pathak, Gueguinou et al. demonstrated how NCLX can affect colorectal cancer progression. It suggests that it may have opposing effects during early and late-stage colorectal cancer, encouraging tumor growth but also decreasing the spread to other organs. Further research could help refine treatments at different stages of the disease.

*2016*). One critical function of mitochondria that is poorly understood in cancer cells is the role of these organelles as a major hub for cellular $Ca^{2+}$ signaling (*De Stefani et al., 2016*; *Pathak and Trebak, 2018*). Essentially all mitochondrial functions are controlled by changes in mitochondrial $Ca^{2+}$ ($mtCa^{2+}$) levels. Increased mitochondrial $Ca^{2+}$ uptake stimulates mitochondrial bioenergetics through the activation of $Ca^{2+}$-dependent dehydrogenases of the tricarboxylic acid (TCA) cycle in the mitochondrial matrix (*Hansford, 1994*; *McCormack et al., 1990*; *Montero et al., 2000*). In turn, $mtCa^{2+}$-mediated changes in mitochondrial metabolism can alter the generation of mitochondrial reactive oxygen species (mtROS). In addition, through changes in $Ca^{2+}$ uptake and extrusion, mitochondria shape the spatial and temporal nature of cytosolic $Ca^{2+}$ signals to regulate downstream gene expression programs (*De Stefani et al., 2016*; *Pathak and Trebak, 2018*).

Mitochondria form very close contact sites with the endoplasmic reticulum (ER) known as mitochondria-associated membranes (*Booth et al., 2016*). These contact sites are hotspots of communication through which $Ca^{2+}$ release from the ER via inositol-1,4,5-trisphosphate receptors ($IP_3R$) is efficiently transferred to the mitochondrial matrix through the mitochondrial $Ca^{2+}$ uniporter (MCU) channel complex (*Baughman et al., 2011*; *Booth et al., 2016*; *De Stefani et al., 2011*; *Wu et al., 2018*). This ensures that the bioenergetic output of the cell is tailored to the strength of cell stimulation by growth factors (*Pathak and Trebak, 2018*). The major $mtCa^{2+}$ extrusion route is mediated by the mitochondrial $Na^+/Ca^{2+}/Li^+$ exchanger (NCLX) (*Palty et al., 2010*). The balance between $Ca^{2+}$ uptake by MCU and $Ca^{2+}$ extrusion by NCLX is critical for maintaining $mtCa^{2+}$ homeostasis, which in turn regulates metabolism and cell fate. Perturbations in $mtCa^{2+}$ homeostasis have been linked to a multitude of diseases including cancer (*De Stefani et al., 2016*; *Pathak and Trebak, 2018*), and altered $mtCa^{2+}$ homeostasis has recently emerged as a novel hallmark of cancer cells (*Danese et al., 2017*; *Kerkhofs et al., 2018*; *Paupe and Prudent, 2018*). However, the underlying molecular mechanisms by which $mtCa^{2+}$ regulate cancer progression are still poorly understood.

In recent studies, altered MCU expression has been reported in cancer cells, and increased expression of MCU and alterations in proteins that regulate MCU channel activity have been associated with increased $mtCa^{2+}$ and downstream effects that contribute to proliferation and tumor progression (*Koval et al., 2019*; *Marchi et al., 2013*; *Ren et al., 2017*; *Tosatto et al., 2016*; *Vultur et al., 2018*). In comparison to MCU, NCLX activity is ~100 fold slower, thus NCLX operation is the rate-limiting factor in $mtCa^{2+}$ homeostasis (*Ben-Kasus Nissim et al., 2017*; *Palty et al., 2010*). For instance, cardiomyocyte-specific MCU knockout mice have unaltered levels of mitochondrial matrix $Ca^{2+}$ and no obvious phenotype, whereas cardiomyocyte-specific NCLX knockout mice

display mtCa$^{2+}$ overload and die from sudden cardiac arrest (*Luongo et al., 2017*). This suggests that although lack of MCU is compensated for by an alternative pathway, the lack of NCLX is not (*Pathak and Trebak, 2018*). Given the importance of NCLX as the major extrusion mediator of mitochondrial matrix Ca$^{2+}$, it is imperative to understand how aberrant expression of this protein influences mitochondrial and cellular function in cancer. However, to date, the role of NCLX in tumor biology has not been directly investigated.

Colorectal cancer (CRC) is the third most commonly diagnosed cancer type and the third leading cause of cancer deaths in both men and women in the United States (*Siegel et al., 2020*). Patient diagnosis at an advanced stage with significant metastatic spread is associated with a 5 year overall survival rate in less than 15% of these CRC patients, demonstrating a need to further understand the underlying mechanisms driving CRC metastasis, recurrence, and development of chemoresistance (*Siegel et al., 2020*). In order to better classify and inform therapeutic interventions the CRC Subtyping Consortium derived four CRC consensus molecular subtypes (CMS) based on comprehensive molecular signatures including mutation, DNA copy number alteration, DNA methylation, microRNA, and proteomics data. These four subtypes differ in their metastatic potential and in survival outcomes of patients (*Guinney et al., 2015*). For example, CMS4, which is characterized as mesenchymal, is associated with an epithelial-to-mesenchymal transition (EMT) phenotype and poor patient survival statistics. In addition, colorectal cancer cell-intrinsic transcriptional signatures (CRIS), which exclude the contribution of tumor-associated stroma, similarly demonstrate the existence of distinct subtypes based on transcriptional profiling (*Dunne et al., 2017*; *Isella et al., 2017*).

Here, we show that the expression of NCLX is significantly downregulated in human colorectal adenocarcinomas. We demonstrate that a loss of NCLX decreases mtCa$^{2+}$ extrusion in CRC cells and that this increase in mtCa$^{2+}$ has important consequences on colorectal tumor cells: NCLX loss (1) inhibits proliferation and primary tumor growth, while (2) enhances metastasis, and drug resistance, suggesting that a loss of NCLX contributes to CRC metastatic progression. Importantly, decreased NCLX expression leads to transcriptional changes reminiscent of highly metastatic mesenchymal CMS4 CRC subtype, including increased expression of genes regulating EMT and cancer stemness, and decreased expression of cell-cycle progression mediators. Mechanistically, decreased expression or loss of NCLX results in mtCa$^{2+}$ overload, causing depolarization of mitochondria, increased mtROS production, which drives ROS-dependent HIF1$\alpha$ protein stabilization and pro-metastatic phenotypes of NCLX-low CRC cells. Thus, we show a novel dichotomous role of NCLX in cancer, where reduced NCLX function lead to reduced tumor growth, while driving a mesenchymal phenotype that leads to increased metastasis and drug resistance.

## Results

### NCLX expression is reduced in colorectal tumors

Using the publicly available Cancer Genome Atlas (TCGA) database, we found that NCLX (*SLC8B1*) mRNA levels were significantly downregulated in both colon and rectal adenocarcinoma (COAD-READ) tumors as compared to the adjacent normal tissue (*Figure 1A*). Consistent with the TCGA data, we observed a substantial reduction to a near loss in *SLC8B1* mRNA in colorectal tumor samples isolated from patients undergoing surgery at Penn State University Medical Center as compared to the paired normal adjacent tissues (*Figure 1B*). There was no difference in *SLC8B1* mRNA levels between male and female tumor tissues (*Figure 1—figure supplement 1A*) and between patients bearing mutated and normal proto-oncogene KRAS and phosphoinositide 3-kinases (PI3K) (*Figure 1C*). However, low NCLX expression was associated with TP53 mutations and wild type BRAF tumors (*Figure 1D*). TCGA data analysis from UALCAN (*Chandrashekar et al., 2017*) revealed that *SLC8B1* mRNA was appreciably reduced in CRC patients of all age groups (*Figure 1—figure supplement 1B*). Both adenocarcinoma and mucinous adenocarcinoma had a significant reduction in *SLC8B1* mRNA levels as compared to the normal tissue (*Figure 1—figure supplement 1C*). Subsequent analysis revealed a significant loss of NCLX in adenomas with malignant transformation from stage I through stage IV (*Figure 1E*). There was a significant reduction in *SLC8B1* mRNA level in late-stage (stage III and IV) colorectal tumors as compared to early-stage (stages I and II) tumors from the TCGA database (*Figure 1E,F*), with similar results when we analyzed the patient samples obtained from Penn State University Medical Center (*Figure 1G*). Together, these results show that

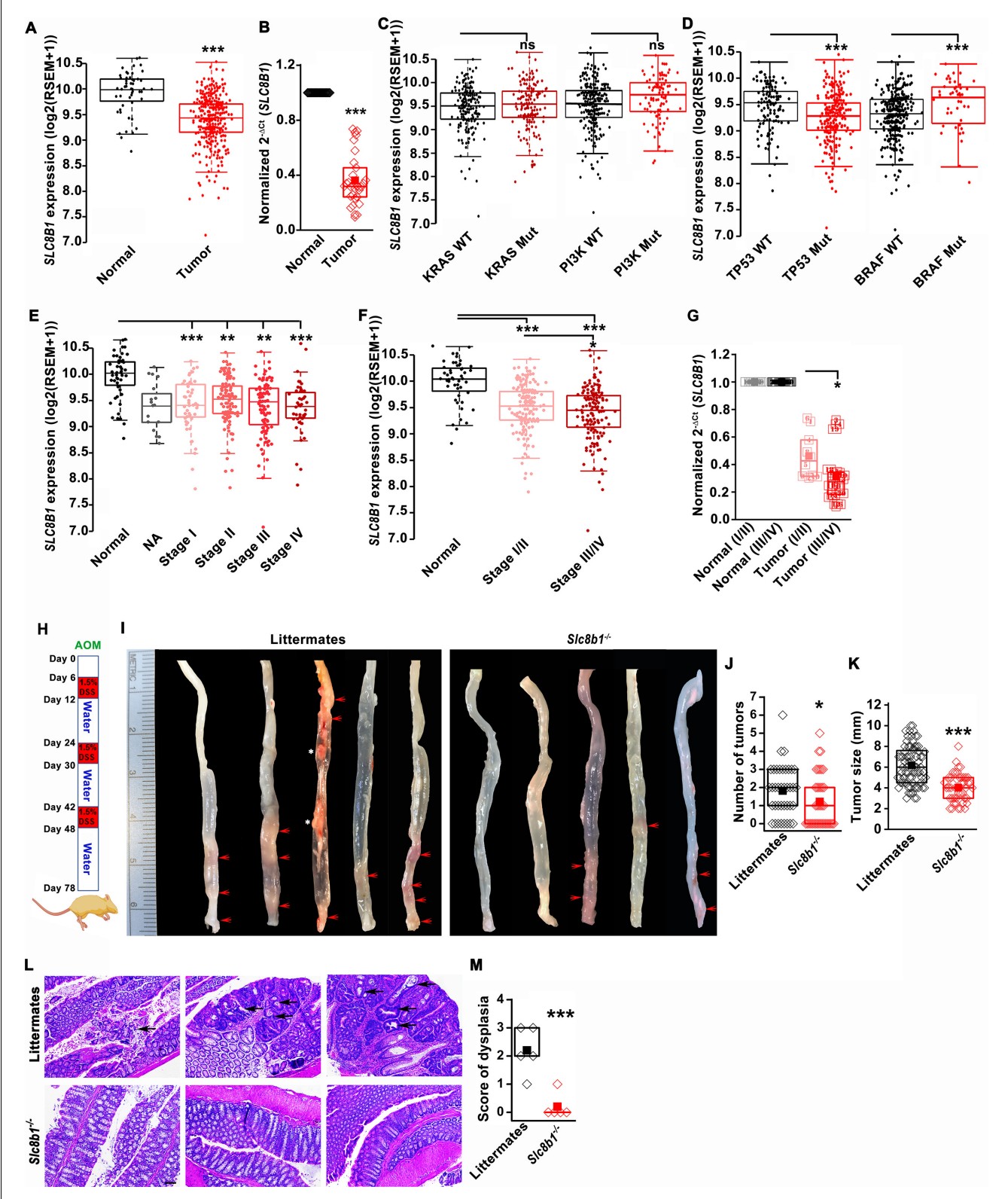

**Figure 1.** The expression of NCLX, a mtCa$^{2+}$ extrusion mediator in CRC cells, is decreased in CRC tumor samples from human patients. (**A**) TCGA data analysis showing *SLC8B1* mRNA levels in tumor tissues and adjacent normal tissues of COADREAD (colon and rectal adenocarcinoma) patients. Each data point represents an individual sample. (**B**) RT-qPCR analysis of *SLC8B1* mRNA in tumor tissues (n = 30) and adjacent normal tissues (n = 30) of CRC patients from Penn State University Hospital. (**C, D**) TCGA data analysis showing *SLC8B1* mRNA level in patients with and without KRAS, PI3K, (**C**) TP53,

*Figure 1 continued on next page*

*Figure 1 continued*

and BRAF (D) mutation. (E–F) TCGA data analysis showing NCLX mRNA in tumors at different cancer stages (stages I–IV) (E) or combined stage I/II (early stage) and stage III/IV (late-stage) (F) of COADREAD tissues compared to adjacent normal tissues. NA = stage not known (G) RT-qPCR analysis of *SLC8B1* mRNA in combined stage I/II (n = 9) and stage III/IV (n = 20) CRC tumor samples compared to their adjacent normal tissues obtained from Penn State University hospital. (H) Schematic representation of the colitis-associated regimen of AOM and DSS treatment. (I–K) Five representative colons from each experimental group are shown (I), quantification of the number of tumors (J), and tumor volume (K) in NCLX KO and control littermate mice at day 78 after AOM/DSS treatment. The red arrow indicates polyps in the colon and the white star represents fat tissue; n $\geq$ 30 mice per group. (L, M) Three replicates of representative H and E staining of colon sections where black arrows indicate dysplasia (scale bar 500 μm) (L), histology score of dysplasia (scale bar 500 μm) (M). Kruskal-Wallis ANOVA was performed to test single variables between the two groups. *p<0.05, **p<0.01, and ***p<0.001.

The online version of this article includes the following source data and figure supplement(s) for figure 1:

**Source data 1.** Source data *Figure 1*.
**Figure supplement 1.** Loss of NCLX reduces tumor number and size in the colitis-associated cancer model.

NCLX expression is significantly downregulated in CRC specimens, and that NCLX loss correlates with late-stage colorectal adenocarcinomas.

## Loss of NCLX has a dichotomous role on tumor growth and metastasis in vivo

To determine the functional link between loss of NCLX expression and the development of CRC in vivo, we first used global NCLX (*Slc8b1*) KO (NCLX KO) mice that were generated using CRISPR/Cas9, where 13 nucleotides (120513241–120513253 on chromosome 13, a region coding for five amino acids) were deleted from the first exon of *Slc8b1*, resulting in a frameshift mutation and an early stop codon in exon 2 at nucleotide 248 of the coding sequence (*Figure 1—figure supplement 1D*). This deletion was confirmed by genotyping of genomic DNA using specific primers for wildtype and knockout alleles (*Figure 1—figure supplement 1E*). We then used the NCLX KO mice and their wildtype littermate controls to determine the contribution of NCLX to the development of colorectal tumors in the colitis-associated colorectal cancer model. We subjected NCLX KO mice (n $\geq$ 30) and littermate control mice (n $\geq$ 30) to one intraperitoneal injection of azoxymethane (AOM; 100 μl of 1 mg/ml) and three cycles of dextran sodium sulfate (DSS; 1.5%) in drinking water with two weeks of normal water between each DSS cycle (*Figure 1H*). At day 78, mice were sacrificed, and their colorectal tracts were harvested. Colorectal tissues from five representative mice from each experimental group are shown in (*Figure 1I*), and the tissues from fifteen additional mice are depicted in (*Figure 1—figure supplement 1F,G*). Although there is a clear association between NCLX loss and CRC in TCGA data, the colons of NCLX KO mice displayed approximately 50% less tumors than those of littermate control mice (*Figure 1I,J* and *Figure 1—figure supplement 1F,G*). Further, the tumors that developed in the colons of NCLX KO mice were markedly smaller than those in the colons of littermate control mice, as determined by measurements of tumor size (*Figure 1K*). Histological analyses of colon tissues revealed significantly reduced dysplasia in the colons of NCLX KO mice compared with colons from littermate control mice (*Figure 1L,M*).

The DSS/AOM administration in mice is an excellent tumor growth model that is not associated with tumor metastases. Therefore, to further investigate the role of NCLX on CRC tumor growth and metastatic spread in vivo, we utilized the human CRC parental cell line HCT116 and its NCLX knockout counterpart in an intrasplenic xenograft model. While the overall efficiency of liver metastasis and colonization is low during intrasplenic injection, this model remains nonetheless is a well established method to study metastasis to distant organs such as the colon and liver (*Heijstek et al., 2005*). For in vivo studies NCLX KO clone #33 was used, which was generated using a guide RNA (g2) resulting in a single cut at nucleotide 150 in exon one causing a frameshift mutation and introduction of a stop codon at predicted position 181-183 bp and 184-186 bp in the NCLX open reading frame (*Figure 2A*). The HCT116 NCLX KO cells and their control HCT116 counterparts were tagged with luciferase and injected (5 $\times$ 10$^5$ cells/mouse) in the spleens of two groups of NOD-SCID mice for a total of 15 mice per experimental group (*Figure 2A*). In vivo metastasis to the colon and liver was assessed by monitoring luciferase bioluminescence. The mice were injected IP with 100 μl luciferin, and the total flux was measured using the In Vivo Imaging System (IVIS) by exposing the mice for 2 min. There was a significant reduction in the total luciferase bioluminescence flux in mice

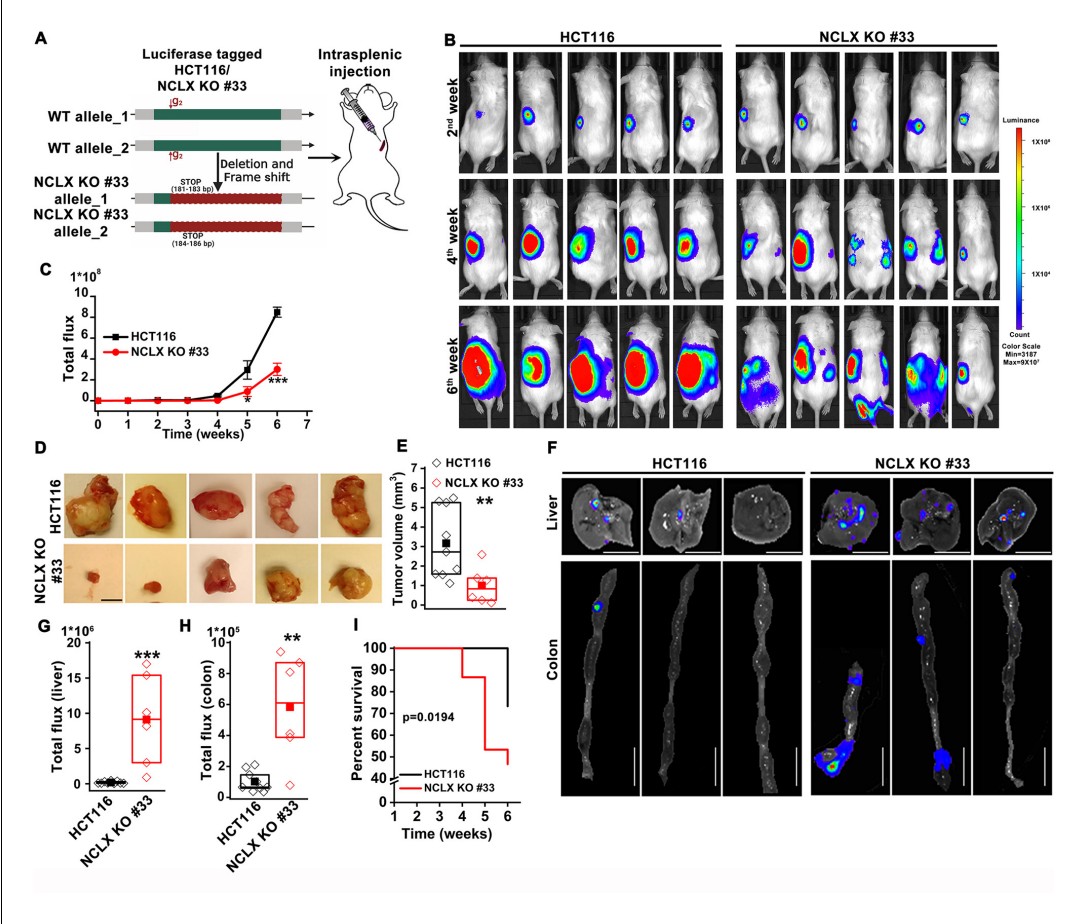

**Figure 2.** Loss of NCLX has a dichotomous role in tumor growth and metastasis in vivo. (A) Schematic representation of CRISPR-generated HCT116 NCLX KO #33 cells. Luciferase-tagged control HCT116 and HCT116 NCLX KO #33 cells were injected at $5 \times 10^5$ cells/mice into spleens of male NOD-SCID mice. (B–C) Representative bioluminescence images (five mice per group) of cancer progression and metastasis in male NOD-SCID mice injected with luciferase-tagged HCT116 cells or HCT116 NCLX KO #33 cells (B) and quantification of whole-body luciferase count (C). (n = 15 mice per group). (D, E) Representative images of the primary tumors at the site of injection (D) and quantification of primary tumor volume at the time of sacrifice (E). Scale bar, 5 mm. (F–H) Representative image of the liver (scale bar, 5 mm) and corresponding colon (scale bar, 1 cm) (F), quantification of luciferase count from the liver (G) and the colon (H) from NOD-SCID mice injected with HCT116 cells or HCT16 NCLX KO #33 cells. (I) Survival curve of NOD-SCID mice injected with either HCT116 cells or HCT116 NCLX KO #33 cells (*p<0.0194, n = 15 mice per group). Kruskal-Wallis ANOVA was performed to test single variables between the two groups. *p<0.05, **p<0.01, and ***p<0.001.

The online version of this article includes the following source data and figure supplement(s) for figure 2:

**Source data 1.** Source data *Figure 2*.

**Figure supplement 1.** Xenografts of HCT116 cells and HCT116 NCLX KO #33 cells in NOD/SCID.

xenografted with HCT116 NCLX KO cells by comparison to the HCT116 control-injected mice (*Figure 2B,C*), indicating that the loss of NCLX in CRC cells caused reduced tumor growth. The luciferase flux at 2, 4, and 6 weeks from five representative mice per experimental group are shown in (*Figure 2B*) with the remaining 10 mice represented in (*Figure 2—figure supplement 1*). Similarly, the primary tumor volumes (at the time of sacrifice, at six weeks) in the spleens of HCT116 NCLX KO-injected mice were significantly reduced compared to control HCT116-injected mice (*Figure 2D, E*), consistent with the in vivo tumor growth in the AOM-DSS model. Interestingly, SCID mice injected with the HCT116 NCLX KO cells showed strikingly increased metastasis (*Figure 2B*), specifically to the liver and colon as compared to control HCT116-injected mice at the time of sacrifice (Week 6; *Figure 2F–H*). We did not observe any metastasis to the lung, heart or brain of mice of both groups in this xenograft model. Significantly, the SCID mice with intra-splenic injection of HCT116 NCLX KO cells had reduced overall survival compared to mice injected with control

HCT116 cells (*Figure 2I*), suggesting that increased CRC metastasis is the primary cause of lethality in the HCT116 NCLX KO xenograft model. Altogether, our results show that loss of NCLX causes reduced primary tumor growth with increased metastatic progression of colorectal cancer.

## Loss of NCLX inhibits proliferation but enhances migration and invasion of CRC cells

To elucidate the mechanisms by which loss of NCLX elicits these seemingly dichotomous functions on CRC, we investigated the effects of CRISPR/Cas9-mediated knockout, and si/shRNA-driven decreases in NCLX expression in HCT116 and DLD1 CRC cell lines (see Methods and *Figure 3—figure supplement 1A–H*). To alleviate potential off-target effects of the CRISPR/Cas9 system, we generated several independent clones obtained with three independent guide RNAs (gRNAs; see methods) (*Figure 3—figure supplement 1A–H*). Genome sequencing and PCR on genomic DNA confirmed NCLX KO (*Figure 3—figure supplement 1G*). With the exception of one commercially available polyclonal NCLX antibody (*Ben-Kasus Nissim et al., 2017*) that is now discontinued, there are currently no reliable NCLX antibodies. All commercially available NCLX antibodies have failed our validation assays and our own attempts to generate a monoclonal antibody against NCLX have not yet produced a reliable clone that detects native levels of NCLX expression. We thus resorted to mRNA quantification, and in all three clones of HCT116 NCLX KO cells, RT-qPCR showed complete absence of *SLC8B1* mRNA (*Figure 3—figure supplement 1D*). Similarly, NCLX KO #06, 24, and 32 of DLD1 cells all had an almost complete deletion of the NCLX open reading frame (*Figure 3—figure supplement 1E*), which was confirmed by PCR on genomic DNA (*Figure 3—figure supplement 1F,G*) and RT-qPCR quantifying mRNA (*Figure 3—figure supplement 1H*).

Since the loss of NCLX reduced tumor size in both AOM-DSS and xenograft models (*Figures 1* and *2*), we assessed the effect of reduced NCLX function on the proliferation of CRC cells by CyQUANT proliferation assays. A significant reduction in proliferation was observed in NCLX KO clones of both HCT116 and DLD1 cells (*Figure 3A,B*). To rule out the possibility of long-term compensation in NCLX KO clones, we downregulated NCLX in HCT116 cells using two independent shRNAs and validated the downregulation by qPCR, showing around 60% reduction in *SLC8B1* mRNA levels in HCT116 cells (*Figure 3—figure supplement 1I*). Similarly, transient knockdown of NCLX (NCLX KD) using shRNA reduced HCT116 cell proliferation (*Figure 3—figure supplement 1J*). The decrease in NCLX KO cell proliferation was accompanied by appreciable changes in apoptotic cells. We observed significant increase in cleaved caspase-3 protein using immunofluorescence staining in HCT116 NCLX KO clones as compared to control HCT116 cells, suggesting an increase in apoptosis of NCLX KO CRC cells (*Figure 3—figure supplement 1K,L*). These data suggest that the reduced tumor sizes observed in vivo (*Figures 1* and *2*) due to NCLX knockout are likely a consequence of reduced CRC cell proliferation and increased apoptosis.

Given that NCLX knockout increased metastatic spread in xenograft models (*Figure 2*), we investigated the effect of NCLX knockout and knockdown on the migration pattern of CRC cells. A gap closure assay revealed that all NCLX KO clones of both HCT116 cells (*Figure 3C,D*) and DLD1 cells (*Figure 7—figure supplement 1J light colors*) had a marked increase in migration at 24 hr. Similar to knockout, shRNA-mediated knockdown of NCLX in HCT116 cells also caused a significant increase in cell migration at 12 and 24 hr time points (*Figure 3E,F*). Although we showed above that the proliferation of NCLX KO clones of HCT116 cells was inhibited compared to control HCT116 cells (*Figure 3A*), we ruled out any potential contribution from proliferation in the cell migration assays by analyzing the migration of HCT116 cells and the HCT116 NCLX KO #33 cells at the 6 hr time point. At 6 hr, we observed a significant increase in migration of NCLX KO #33 cells as compared to HCT116 control cells (*Figure 3—figure supplement 1M*). Effects of proliferation on cell migration were further ruled out by documenting that the increased cell migration of HCT116 NCLX KO cells is preserved in the presence of the cytostatic compound, mitomycin C at 12 hr and 24 hr time points (*Figure 3—figure supplement 1N*).

Further, the invasive behavior of NCLX KO cells was determined using a Matrigel-coated Boyden chamber assay. A marked increase in invasion was observed in all NCLX KO clones of both HCT116 and DLD1 as compared to their respective controls (*Figure 3G,H*, *Figure 3—figure supplement 1O*). Supporting this invasive phenotype are our observations that mRNA levels of the matrix metalloproteinases MMP1, 2, and 9 were significantly upregulated in all the HCT116 NCLX KO clones, as compared to control HCT116 cells (*Figure 3I–K*). Similarly, the protein levels of MMP1, 2, and 9

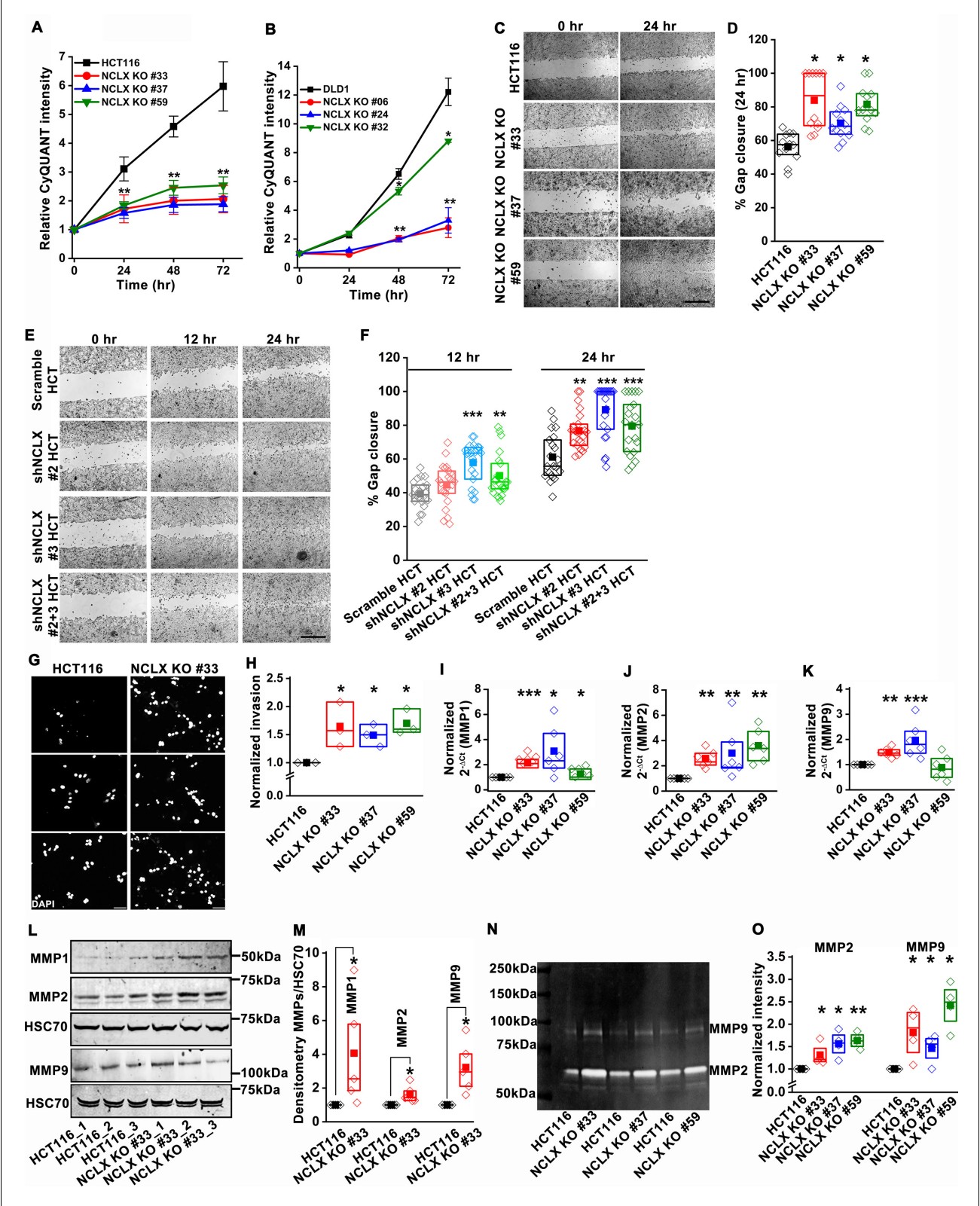

**Figure 3.** Loss of NCLX inhibits proliferation but enhances migration and invasion of CRC cells. (A, B) CyQUANT proliferation assays of HCT116 cells (A), DLD1 cells (B), and their respective NCLX KO clones. (C, D) Representative bright-field images of in vitro wound healing migration assay (C), and quantification of % gap closure for HCT116 cells and clones of HCT116 NCLX KO cells after 24 hr (D). Scale bar, 1 mm. (E, F) In vitro migration assays of control HCT116 cells infected with either scramble shRNA or two different shRNA sequences against NCLX (E) and quantification of % gap closure (F).

*Figure 3 continued on next page*

*Figure 3 continued*

Scale bar, 1 mm. (G, H) Representative images of invasion assays performed in triplicates (G), and quantification of normalized invasion of HCT116 cells and clones of HCT116 NCLX KO cells (H). Scale bar, 50 µm. (I–K) RT-qPCR data showing mRNA expression for MMP1 (I), MMP2 (J), and MMP9 (K) in HCT116 cells and clones of HCT116 NCLX KO cells. (L, M) Western blot probed with anti-MMP1, anti-MMP2, anti-MMP9, and anti-HSC70 antibody as a loading control (L), and quantification of MMPs band intensities normalized to that of HSC70 (M). (N, O) Representative gelatin zymogram showing MMP2 and MMP9 activity from HCT116 cells, and their respective clones of HCT116 NCLX KO cells (N), quantification of band intensities of MMP2 and MMP9 activities (O). All experiments were performed ≥three times with similar results. Statistical significance was calculated using one-way ANOVA followed by a post-hoc Tukey test, except for M and O, where the paired t-test was performed. *p<0.05, **p<0.01 and ***p<0.001.

The online version of this article includes the following source data and figure supplement(s) for figure 3:

**Source data 1.** Source data *Figure 3*.
**Figure supplement 1.** Deletion of NCLX results in reduced proliferation and increased migration and invasion.
**Figure supplement 1—source data 1.** Source data *Figure 3—figure supplement 1*.

were also significantly increased in NCLX KO clones of both HCT116 and DLD1 cells (*Figure 3L,M*, *Figure 3—figure supplement 1P,Q*). MMP9 and MMP1 protein levels were slightly increased in HCT116 cells in which NCLX was knocked down by shRNA (NCLX KD), although for MMP1, this increase was not statistically significant (*Figure 3—figure supplement 1R,S*). We then tested the MMPs activity using zymography and revealed that the activity of MMP2 and 9 were markedly increased in all the NCLX KO clones of HCT116 cells compared to control HCT116 cells (*Figure 3N, O*). This was more prominent than changes observed at the mRNA and protein levels, suggesting that the loss of NCLX expression in CRC cells mostly contributes to post-transcriptional regulation of MMPs. Collectively, these results suggest that the loss of NCLX in CRC cells causes an increase in migration and invasion of CRC cells through increased MMP1, 2, and 9 protein levels and activity. The data also confirm the dichotomous role of NCLX knockout observed in vivo, demonstrating that NCLX loss results in decreased cell proliferation and an increase in migratory and invasive phenotypes.

## Loss of NCLX in CRC cells inhibits mtCa²⁺ extrusion, causes mitochondrial perturbations, and enhances mitochondrial ROS

We and others have previously shown that NCLX is the major molecular mediator of mtCa$^{2+}$ extrusion and that inhibiting NCLX expression and function prevents mtCa$^{2+}$ extrusion (*Ben-Kasus Nissim et al., 2017*; *Palty et al., 2010*). We measured mtCa$^{2+}$ extrusion in all NCLX KO clones by loading the cells with the mitochondrial Ca$^{2+}$ sensitive dye Rhod-2 AM. Cells were co-loaded with the mitochondrial specific dye MitoTracker Green FM as a control. Cells were then stimulated with ATP, a purinergic G protein-coupled receptor (P2Y) agonist that couples to phospholipase Cβ (PLCβ) activation and subsequent inositol-1,4,5-trisphosphate (IP$_3$)-dependent release of Ca$^{2+}$ from the ER through IP$_3$ receptors (*Buvinic et al., 2009*; *Gonzalez et al., 1989*). Upon stimulation with 300 µM ATP in the presence of extracellular Ca$^{2+}$, a portion of Ca$^{2+}$ released from the ER through IP$_3$ receptors is transferred to mitochondria. Hence, cells showed a biphasic response with an increase in Rhod-2 fluorescence followed by a decrease in fluorescence, corresponding to mtCa$^{2+}$ uptake and mtCa$^{2+}$ extrusion, respectively. All NCLX KO clones showed a significant reduction in mtCa$^{2+}$ extrusion (*Figure 4—figure supplement 1A–L*) with no significant change in the rate of mtCa$^{2+}$ uptake. To rule out the possibility of long-term compensation in NCLX KO clones, we transiently downregulated NCLX expression in HCT116 and DLD1 cells using siRNA and validated NCLX knockdown by RT-qPCR. We observed around ~60% reduction in *SLC8B1* mRNA levels in both HCT116, DLD1, and HT29 cells (*Figure 4—figure supplement 1M*). Similar to the NCLX KO cells, HCT116 cells, DLD1 cells, and HT29 cells transfected with siRNA against NCLX (siNCLX) exhibited a significant reduction in mtCa$^{2+}$ extrusion with no significant change in mtCa$^{2+}$ uptake (*Figure 4—figure supplement 1N–V*). We previously showed that inhibition of mtCa$^{2+}$ extrusion through NCLX knockdown in several cell types leads to the inhibition of plasma membrane ORAI1 channels, reduced Ca$^{2+}$ entry from the extracellular space, and decreased cytosolic Ca$^{2+}$ (*Ben-Kasus Nissim et al., 2017*). Therefore, we measured cytosolic Ca$^{2+}$ in response to stimulation with ATP in HCT116 cells and their NCLX KO #33 counterparts using the dye Fura-2 and showed a reduction in Ca$^{2+}$ entry in NCLX KO cells with no effect on Ca$^{2+}$ release from the ER (*Figure 4—figure supplement 1W,X*). Collectively, these results show that inhibition of NCLX function in CRC cell lines leads to enhanced mitochondrial

matrix $Ca^{2+}$ concentration due to reduced $mtCa^{2+}$ extrusion and to decreased cytosolic $Ca^{2+}$ due to reduced $Ca^{2+}$ entry across the plasma membrane.

The decrease in $mtCa^{2+}$ extrusion in HCT116 and DLD1 cell clones in which NCLX expression was either reduced by siRNA knockdown or ablated by CRISPR/Cas9 knockout suggested that these clones may be experiencing mitochondrial $Ca^{2+}$ overload. In normal cells, mitochondrial $Ca^{2+}$ overload alters bioenergetics and causes drastic cellular dysfunction leading to cell death (*Celsi et al., 2009*; *Santulli et al., 2015*). This occurs mainly through the opening of the mitochondrial permeability transition pore (mPTP) (*Bernardi and Di Lisa, 2015*; *Halestrap, 2009*) and subsequent mitochondrial membrane depolarization. Therefore, the effects of NCLX knockout on mitochondrial membrane potential were measured using the tetramethylrhodamine methyl ester (TMRE) dye. We observed a significant decrease in the accumulation of TMRE in NCLX KO cells, indicating that mitochondria of NCLX KO cells are more depolarized than control HCT116 and DLD1 cells (*Figure 4A, B*).

Mitochondrial depolarization is a sign of mitochondrial damage and a major driver of mitophagy, by mediating Pink1 accumulation at the outer mitochondrial membrane (*Jin et al., 2010*). Examining mitochondrial structure using transmission electron microscopy (TEM) imaging revealed that 60–70% of mitochondria in NCLX KO CRC clones showed altered shape and disrupted cristae compared to control cells (*Figure 4C* and *Figure 4—figure supplement 2A–C*). We discovered that mitochondrial membranes and cristae of the NCLX KO clones of HCT116 and DLD1 cells were disrupted (*Figure 4C*, *Figure 4—figure supplement 2A–C*). Specifically, while overall mitochondrial area was not changed in NCLX KO clones (*Figure 4D*), we observed a greater number of mitochondria with disordered cristae (*Figure 4E*) and a decrease in the number of intact cristae per mitochondrion (*Figure 4F*). We also observed a significant increase in autophagic vesicles in NCLX KO cells compared to control HCT116 cells (*Figure 4—figure supplement 2D*). The number of mitophagic vesicles were slightly increased in NCLX KO cells compared to control HCT116 cells (e.g., see images in *Figure 4—figure supplement 2C*), although this increase was not statistically significant (*Figure 4—figure supplement 2E*). The autophagy/mitophagy vesicle marker LC3BII was significantly increased in all NCLX KO clones of HCT116 and DLD1 cells (*Figure 4G,H*, and *Figure 4—figure supplement 2F,G*). Interestingly, the levels of the p62 protein, a signaling molecule that is downstream of LC3BII that is degraded on induction of autophagy (*Liu et al., 2016*; *Youle and Narendra, 2011*), were also increased in all the NCLX KO clones of HCT116 and DLD1 cells (*Figure 4G,I* and *Figure 4—figure supplement 2F,G*). Similarly, the shRNA-mediated knockdown of NCLX in HCT116 (see *Figure 3—figure supplement 1I* for evidence of *SLC8B1* mRNA knockdown using shRNA) resulted in increased LC3BII and p62 protein levels (*Figure 4—figure supplement 2H,I*).

To assess if these mitochondrial perturbations have consequences on mitochondrial electron transport chain function, we measured mitochondrial oxygen consumption rate (OCR) of NCLX KO HCT116 and DLD1 cells using Seahorse extracellular flux assays. We observed a significant reduction in maximal respiration and spare respiratory capacity in NCLX KO clones of both HCT116 and DLD1 cell (*Figure 4J–M*, and *Figure 4—figure supplement 2J–M*). Interestingly, basal respiration and ATP generation were significantly reduced in DLD1 NCLX KO clones, and remained mostly unaltered in HCT116 NCLX KO clones as compared to their respective controls (*Figure 4—figure supplement 2J–M*, and *Figure 4J–M*). We determined whether reduced OCR in NCLX KO clones was due to altered protein expression of the mitochondrial respiratory complexes I-V. To access the changes in mitochondrial respiratory complexes, we measured the protein levels of one component from each of the five complexes including NADH dehydrogenase [ubiquinone] 1 β subcomplex subunit 8 (NDUFB8, Complex-I), Succinate dehydrogenase [ubiquinone] iron-sulfur subunit (SDHB, Complex-II), Cytochrome b-c1 complex subunit 2 (UQCRC2, Complex-III), Cytochrome c oxidase subunit 1 (MTCO1, Complex IV), and ATP synthase subunit α (ATP5A, Complex-V). Interestingly, we did not observe any significant change in the expression of any of the above described components of the mitochondrial respiratory Complexes I–V between HCT116 cells and their NCLX KO clones (*Figure 4—figure supplement 2N,O*). These data suggest that lack of NCLX does not completely abrogate basal mitochondrial respiration, but negatively affects respiratory reserve, which is an indication of the cells ability to enhance mitochondrial respiration in response to higher energy demands.

A further consequence of $mtCa^{2+}$ overload is the generation of mitochondrial reactive oxygen species (mtROS) (*Bertero and Maack, 2018*; *Brookes et al., 2004*). In normal cells, enhanced mtROS can lead to mitochondrial dysfunction and cell death; however, many tumor cells utilize

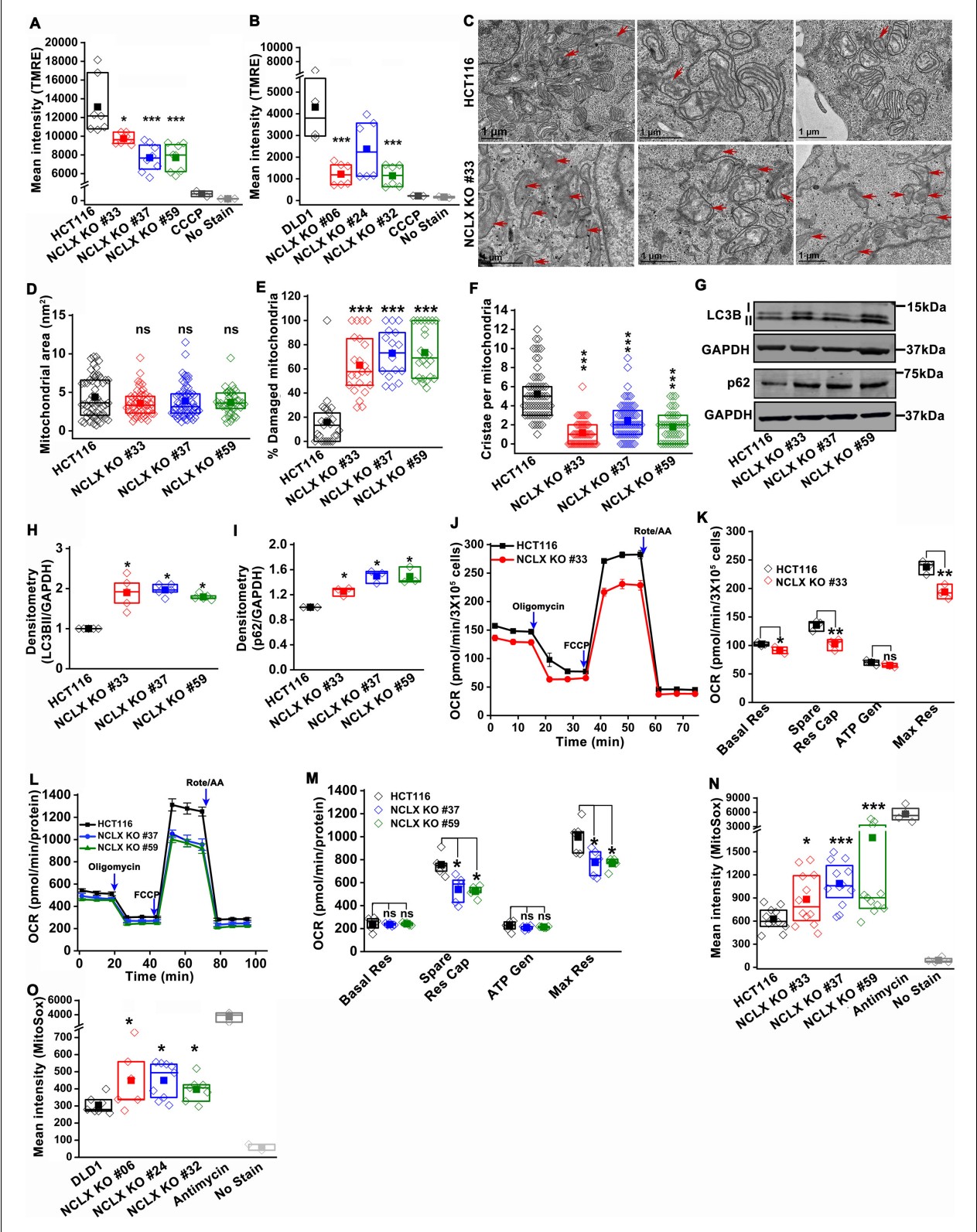

**Figure 4.** Abrogation of NCLX function in CRC cells causes mitochondrial damage, and enhances mitochondrial ROS. (**A**, **B**) Flow cytometric analysis of mitochondrial membrane potential using the dye TMRE in HCT116 cells (**A**) and DLD1 cells (**B**) and their respective NCLX KO clones. Each data point represents one experimental replicate. 100 µM CCCP was added to cells as a positive control, and unstained cells were used as a negative control. (**C**) Representative images of a transmission electron micrograph of HCT116 cells and HCT116 NCLX KO cells. Red arrows indicate damaged mitochondria. *Figure 4 continued on next page*

*Figure 4 continued*

(D–F) Quantification of mitochondrial area (D), number of damaged mitochondria (E), and number of cristae per mitochondria (F) in HCT116 cells and clones of HCT116 NCLX KO cells. (G–I) Western blots labeled with indicated antibodies with anti-GAPDH used as a loading control (G) and quantification of band intensity of LC3B II (H) and p62 (I) normalized to GAPDH in HCT116 cells and clones of HCT116 NCLX KO cells. (J–M) Oxygen consumption rate (OCR) of HCT116 cells and HCT116 NCLX KO clones (J,L), and quantification of basal respiration (Basal Res), spare respiratory capacity (Spare Res Cap), ATP generation (ATP Gen), and maximum respiration (Maximal Res) (K, M). (N, O) Flow cytometric analysis of mitochondrial ROS levels measured with the dye MitoSox in HCT116 cells (N) and DLD1 cells (O) and their respective NCLX KO clones. As a positive control, 50 µM antimycin was used and unstained cells were used as a negative control. All experiments were performed ≥three times with similar results. Statistical significance was calculated using one-way ANOVA followed by a post-hock Tukey test, except figure H, and I, where paired t-test was used. $*p<0.05$, $**p<0.01$, and $***p<0.001$.

The online version of this article includes the following source data and figure supplement(s) for figure 4:

**Source data 1.** Source data *Figure 4*.
**Figure supplement 1.** Loss of NCLX in CRC cells inhibits mtCa$^{2+}$ extrusion.
**Figure supplement 1—source data 1.** Source Data *Figure 4—figure supplement 1*.
**Figure supplement 1—source data 2.** Source Data *Figure 4—figure supplement 2*.
**Figure supplement 2.** Loss of NCLX causes mitochondrial damage.

mitochondria-derived ROS as cellular signals to drive pro-survival adaptations, including changes in downstream transcription (*Vyas et al., 2016*). Hence, we measured mtROS in HCT116 and DLD1 CRC cells and their respective NCLX KO clones using the dye MitoSOX and flow cytometry. Mito-SOX dye intensity was significantly increased in all the NCLX KO clones of both HCT116 and DLD1 cells (*Figure 4N,O*), indicating increased mtROS in these cells. Using fluorescence microscopy, we also show that downregulation of NCLX with siRNA in HCT116 and DLD1 cells (and in another CRC cell line, HT29; See *Figure 4—figure supplement 1M* for evidence of *SLC8B1* mRNA knockdown in HT29) results in a significant increase in mtROS levels (*Figure 4—figure supplement 2P,Q*). The above data demonstrate that loss of NCLX decreases mtCa$^{2+}$ extrusion, which affects mitochondrial cristae morphology and depolarization of the mitochondrial membrane, leading to formation of autophagic vesicles and possibly mitophagy, decreased respiratory reserve capacity, and enhanced mtROS production.

## Loss of NCLX leads to pro-metastatic transcriptional reprogramming

To further delineate the role of NCLX loss in CRC, we performed transcriptional profiling of HCT116 cells and their NCLX KO counterparts using RNA sequencing. Gene Set Enrichment Analysis (GSEA) revealed that NCLX knockout drives the positive enrichment of pathways involved in hypoxia, epithelial-to-mesenchymal transition (EMT), TGF-β, pro-inflammatory, glycolysis, apoptosis and angiogenesis pathways, while negatively influencing gene expression of Myc targets, cell-cycle regulation, and oxidative phosphorylation (*Figure 5A–G*, and *Figure 5—figure supplement 1A–F*). Thus, GSEA analysis revealed that NCLX loss drives gene expression signatures associated with metastatic progression and inhibition of proliferation, mirroring the phenotypic changes we observed following NCLX knockout and knockdown. Interestingly, these gene expression signatures are shared by the mesenchymal CMS4 CRC subtype, which is characterized by high rates of recurrence, and predictive of poor patient outcome (*Guinney et al., 2015*).

## NCLX deficiency causes stem cell-like phenotype and chemoresistance of CRC cells

A phenotype of mesenchymal CRC is the enrichment of cancer stem cell traits (*Guinney et al., 2015*; *Polyak and Weinberg, 2009*). In agreement with these findings, we found that transcript levels of stem cell markers NANOG, Oct4, Sox2, and Foxo3, as well as regulators of the glutathione synthesis pathway implicated in regulating these transcription factors in breast cancer, SLC7A11 and GCLM (*Lu et al., 2015*), were significantly upregulated in NCLX KO cells (*Figure 5H–J*, *Figure 5—figure supplement 1G–O*). One exception was the mRNA levels of Oct4 in DLD1 NCLX KO clones, which were not significantly different from those of control DLD1 cells (*Figure 5—figure supplement 1N*). These data suggest that NCLX KO clones acquire stem cell-like properties.

In most cancer types, a stem cell-like phenotype is associated with enhanced invasion and chemoresistance (*Blank et al., 2018*; *Munro et al., 2018*; *Reya et al., 2001*; *Touil et al., 2014*). The

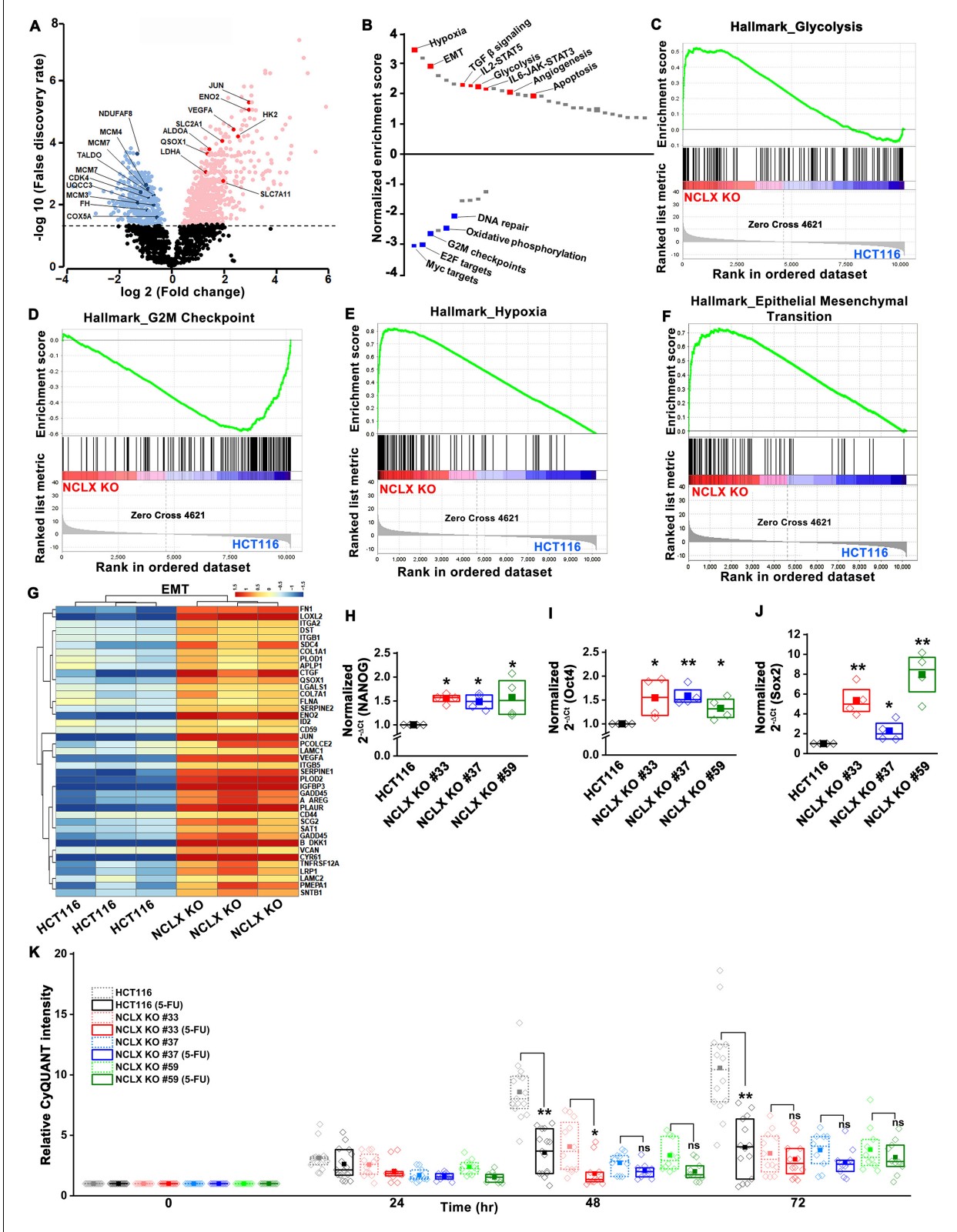

**Figure 5.** Loss of NCLX leads to pro-metastatic transcriptional reprogramming. (**A**) Volcano plot of differentially expressed genes between HCT116 cells and HCT116 NCLX KO #33 cells showing Log2 fold change vs. false discovery rate. The thresholds in the figure correspond to a false discovery rate <0.05 and log2 fold change <−1.5 or>1.5. Genes that are significantly upregulated and downregulated are represented in pink and light blue, respectively. (**B**) Pathway analysis showing normalized enrichment score. Positive enrichment score shows upregulated pathways, and negative

*Figure 5 continued on next page*

*Figure 5 continued*

enrichment score depicts downregulated pathways. (C–F) GSEA analysis of HCT116 cells and HCT116 NCLX KO #33 cells show a positive correlation in the enrichment of hallmark of glycolysis genes (C), a negative correlation of hallmark of G2M checkpoint genes (D), a positive correlation of hallmark of hypoxia-related genes (E), and hallmarks of epithelial-to-mesenchymal transition genes (F). (G) The heat map shows significantly increased expression of EMT genes in HCT116 NCLX KO #33 cells as compared to control HCT116 cells. (H–J) RT-qPCR showing mRNA levels NANOG (H), octamer-binding transcription factor 4 (Oct4) (I), and SRY-Box Transcription Factor 2 (Sox2) (J) in HCT116 cells and clones of HCT116 NCLX KO cells. (K) The proliferation of HCT116 cells and HCT116 NCLX KO #33 cells with and without 10 µM 5-FU treatment. All experiments were performed ≥three times with similar results. Statistical significance was calculated using a paired t-test except for K, where one-way ANOVA followed by a post-hock Tukey test was used. *p<0.05, **p<0.01, and ***p<0.001.

The online version of this article includes the following source data and figure supplement(s) for figure 5:

**Source data 1.** Source data *Figure 5*.
**Figure supplement 1.** Abrogation of NCLX function leads to transcriptional reprogramming.
**Figure supplement 1—source data 1.** Source Data *Figure 5—figure supplement 1*.

chemoresistance properties of the NCLX KO clones and respective control HCT116 and DLD1 cells were tested in response to treatment with the antimetabolite agent 5-Fluorouracil (5-FU), which is widely used in the treatment of CRC. First, we performed titration experiments with doses of 5-FU ranging from 1 to 25 µM. The control HCT116 cells and the HCT116 NCLX KO cells showed a dose-dependent reduction in proliferation in response to 5-FU treatment (*Figure 6—figure supplement 1A*). The $IC_{50}$ of 5-FU for HCT116 NCLX KO cells (15 ± 1.2 µM) was significantly higher than the control HCT116 cells (8.2 ± 1.2 µM) (*Figure 6—figure supplement 1B*). Therefore, subsequent experiments were performed with 10 µM 5-FU. The treatment with 10 µM 5-FU caused a significant reduction in proliferation of control HCT116 and DLD1 cells at 24 and 72 hr (*Figure 5K*, *Figure 6—figure supplement 1C*). Treatment of the NCLX KO clones of HCT116 (*Figure 5K*) and DLD1 (*Figure 6—figure supplement 1C*) cells with 10 µM 5-FU yielded only a marginal reduction in proliferation at 72 hr as compared to non-treated NCLX KO clones, although this reduction was statistically significant for two NCLX KO clones of DLD1 cells at 72 hr time point (Clone#24 and #32; *Figure 6—figure supplement 1C*). 5-FU also caused a significant inhibition in migration of control HCT116 cells (*Figure 6—figure supplement 1D,E*), but did not affect the migratory capabilities of the NCLX KO clones of both HCT116 and DLD1 cells (*Figure 6—figure supplement 1D,E*).

## NCLX deficiency stabilizes HIF1α and regulates migration in a mtROS-dependent manner

The mesenchymal, invasive, and chemoresistance phenotypes of NCLX KO cells, as well the majority of enriched pathways identified by GSEA, including hypoxia, EMT, glycolysis, angiogenesis, and the suppression of OXPHOS, share a common regulator, the hypoxia-inducible factor HIF1α (*Figure 5A–G* and *Figure 5—figure supplement 1A-F*). This was intriguing in light of our observations that NCLX knockdown decreases respiration and increases mitochondrial ROS production in CRC (*Figure 4J–O*), as mtROS is a known activator of HIF1α (*Bell et al., 2007*; *Dan Dunn et al., 2015*; *Hamanaka and Chandel, 2010*; *Pan et al., 2007*). Hence, we measured HIF1α protein levels and found that HIF1α protein levels were strikingly increased in the NCLX KO clones of both HCT116 and DLD1 cells in the absence of a hypoxic stimulus (*Figure 6A–D*). Similarly, HIF1α protein levels were also increased in HCT116 cells in which NCLX was knocked down with shRNA (NCLX KD; *Figure 6E,F*). Importantly, we were able to demonstrate that HIF1α protein stabilization was significantly abrogated in NCLX KO cells when mtROS were scavenged using the mitochondria-targeted antioxidant mitoTEMPO (*Figure 6G,H*). Both AMPK and mTORC1 are known regulators of HIF1α (*Dodd et al., 2015*; *Hudson et al., 2002*; *Tandon et al., 2011*). Therefore, we assessed the phosphorylation levels of AMPK and the ribosomal protein S6K (a readout of mTORC1 activation) in HCT116 and clones of HCT116 NCLX KO cells. Assessment of AMPK and S6K1 phosphorylation revealed no difference between control HCT116 and clones of HCT116 NCLX KO cells (*Figure 6—figure supplement 1F–I*), suggesting that mTORC1 does not contribute to the regulation of HIF1α in response to NCLX ablation. Taken together, these results suggest that the major regulator of HIF1α protein levels in NCLX KO CRC cells is mtROS. Further, a significant reduction in migration was observed when HCT116 NCLX KO cells were treated with mitoTEMPO (*Figure 6I,J*), while this

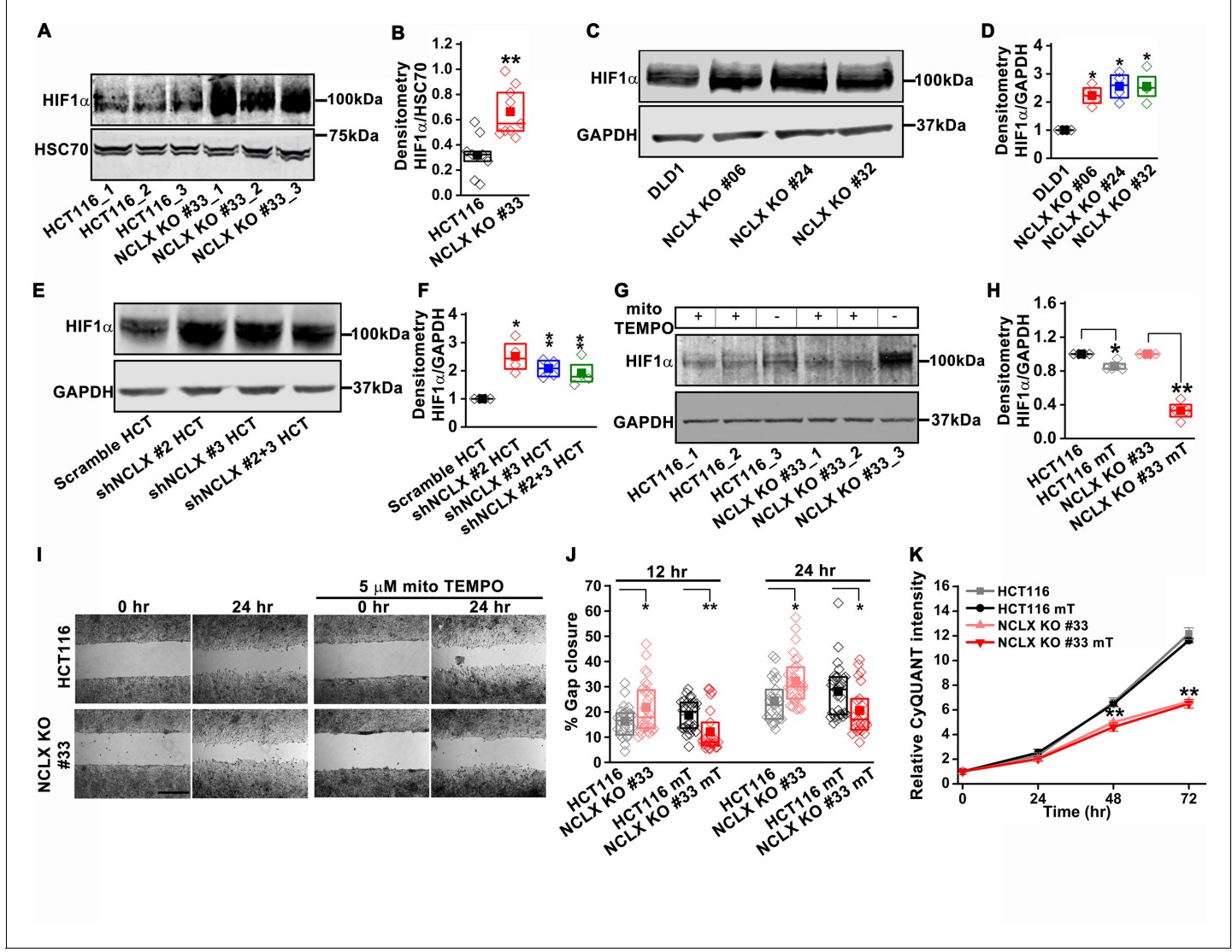

**Figure 6.** NCLX deficiency stabilizes HIF1α and regulates migration in a mtROS-dependent manner. (A–F) Western blot of HCT116 cells and clones of HCT116 NCLX KO cells (A), DLD1 cells and clones of DLD1 NCLX KO cells (C) and HCT116 cells infected with either scramble shRNA or three combinations of shRNA against NCLX (E) probed with anti-HIF1α, anti-GAPDH and anti-HSC70 antibody as a loading control, and quantification of HIF1α band intensity relative to HSC70 or GAPDH (B, D, and F). (G, H) Western blot showing HIF1α proteins in HCT116 cells and clones of HCT116 NCLX KO cells after treatment with 5 μM mitoTEMPO overnight (G), and quantification of HIF1α band intensity normalized to GAPDH (H). (I, J) Representative bright-field images of in vitro migration assay (I), and quantification of % gap closure (J) of HCT116 cells and clones of HCT116 NCLX KO cells, with and without treatment with 5 μM mitoTEMPO. Scale bar, 1 mm. (K) Proliferation of HCT116 cells and HCT116 NCLX KO #33 cells with and without 5 μM mitoTEMPO. All experiments were performed ≥three times with similar results. Statistical significance was calculated using the paired t-test unless mentioned otherwise. *p<0.05, **p<0.01, and ***p<0.001.

The online version of this article includes the following source data and figure supplement(s) for figure 6:

**Source data 1.** Source data *Figure 6*.

**Figure supplement 1.** NCLX KO CRC cells show chemoresistance to 5-FU.

**Figure supplement 1—source data 1.** Source Data *Figure 6—figure supplement 1*.

had no effect on the proliferation of either control HCT116 cells or their NCLX KO counterparts (*Figure 6K*).

## Glycolysis is critical for migration of NCLX-deficient colorectal cancer cells

HIF1α is an important regulator of glycolysis. The transcriptomic analysis (*Figure 5A* and *Figure 5— figure supplement 1A*), GSEA pathway, and enrichment analysis (*Figure 5B,C*) showed that glycolysis-related genes were upregulated in HCT116 NCLX KO cells. RT-qPCR results confirmed that the glycolysis-related genes GLUT1 (glucose transporter one or SLC2A1), HK2 (hexokinase 2), ALDOA (aldolase A), ENO1 (enolase 1), and LDHA (lactate dehydrogenase A) were distinctly upregulated in all the NCLX KO HCT116 clones (*Figure 7A*). Loss of NCLX in either HCT116 or DLD1 increased the protein levels of HK2, ALDOA, and LDHA (*Figure 7B,C*, and *Figure 7—figure supplement 1A–D*). We also validated this further using shRNA-mediated knockdown of NCLX (NCLX KD) in HCT116 cells. *SLC8B1* mRNA levels were reduced by 60–70% in NCLX KD cells compared to cells transfected with shRNA scramble control (*Figure 3—figure supplement 1I*). Similar to NCLX KO, the HCT116 NCLX KD cells showed a modest but significant increase of HK2, ALDOA, and LDHA protein levels (*Figure 7—figure supplement 1E,F*). Interestingly, we observed a significant downregulation of the pentose phosphate pathway genes in HCT116 NCLX KO cells, including decreased mRNA expression of the enzymes glucose 6-phosphate dehydrogenase (G6PD), 6-phosphogluconate dehydrogenase (PGD) and Transketolase (TKT) (*Figure 7—figure supplement 1G*).

Since inhibition of NCLX function caused upregulation of glycolytic genes, we used Seahorse extracellular flux analysis to determine whether NCLX knockout affects the extracellular acidification rate (ECAR) of HCT116 and DLD1 cells. Consistent with the upregulation of glycolytic genes, ECAR, and glucose dependency were increased in all NCLX KO clones (*Figure 7D–F* and *Figure 7—figure supplement 1H*). Furthermore, direct measurements of glucose and lactate in growth media using the YSI system also revealed that the glucose consumption and lactate production were markedly increased in NCLX KO clones of both HCT116 and DLD1 cells (*Figure 7G–J*). These data suggest that NCLX KO cells compensate for their decreased respiratory reserve capacity (*Figure 4J–M*, and *Figure 4—figure supplement 2J–M*) by enhancing glycolytic pathways to meet their energy demands.

Considering these findings, the glycolytic inhibitor 2-deoxy-D-glucose (2-DG) was used to test if CRC cells that lack NCLX expression are more vulnerable to glycolysis inhibition. While 2-DG caused a significant reduction in proliferation of both DLD1 control and NCLX KO cells (*Figure 7—figure supplement 1I*), it did not affect the proliferation of NCLX KO clones of HCT116 (*Figure 7K*). However, the enhanced migration of NCLX KO clones was abrogated when glycolysis was inhibited using 2-DG (*Figure 7L–M*; *Figure 7—figure supplement 1J*), suggesting that increased glycolysis is supporting migration of NCLX KO clones and that targeting this metabolic adaptation may be an avenue to block metastatic progression of CRC cells with low NCLX expression.

## Discussion

In addition to the critical role mitochondria play in cellular bioenergetics, lipid metabolism, and cell death, recent attention has focused on their direct and indirect contributions to mediating crucial signal transduction pathways that control the shift in cellular metabolic activity and changes in gene transcription (*Tennant et al., 2009*; *Tennant et al., 2010*). Mitochondria are generators and active participants in $Ca^{2+}$ and ROS signaling (*Hempel and Trebak, 2017*; *Vyas et al., 2016*). Growing evidence has shown a critical role of $mtCa^{2+}$ homeostasis in mitochondrial function and cell survival (*Celsi et al., 2009*; *Santulli et al., 2015*) and reported a strong correlation between altered mitochondrial function and disease including cancer progression (*Porporato et al., 2018*). Mitochondria are major mediators of the shift in metabolic activity of cancer cells. This metabolic shift is a mechanism of adaptation to stressors within the tumor microenvironment that is required for cancer cell survival (*Vyas et al., 2016*). Nevertheless, the mechanisms of $mtCa^{2+}$ in driving mitochondrial signaling and metabolic shifts have largely remained unknown.

Previous studies on the mitochondrial $Ca^{2+}$ uniporter (MCU) complex indicated that the level of $mtCa^{2+}$ is an important determinant in the progression of different cancer types (*Marchi et al., 2019a*; *Marchi et al., 2013*; *Marchi et al., 2019b*; *Tosatto et al., 2016*). The downregulation of MCU and subsequent reduction of $mtCa^{2+}$ uptake correlated with resistance to apoptosis of colorectal cancer cell lines (*Marchi et al., 2013*). Conversely, in triple-negative breast cancer cell lines, the downregulation of MCU caused a reduction in mtROS, HIF1α levels, cell migration, invasion and

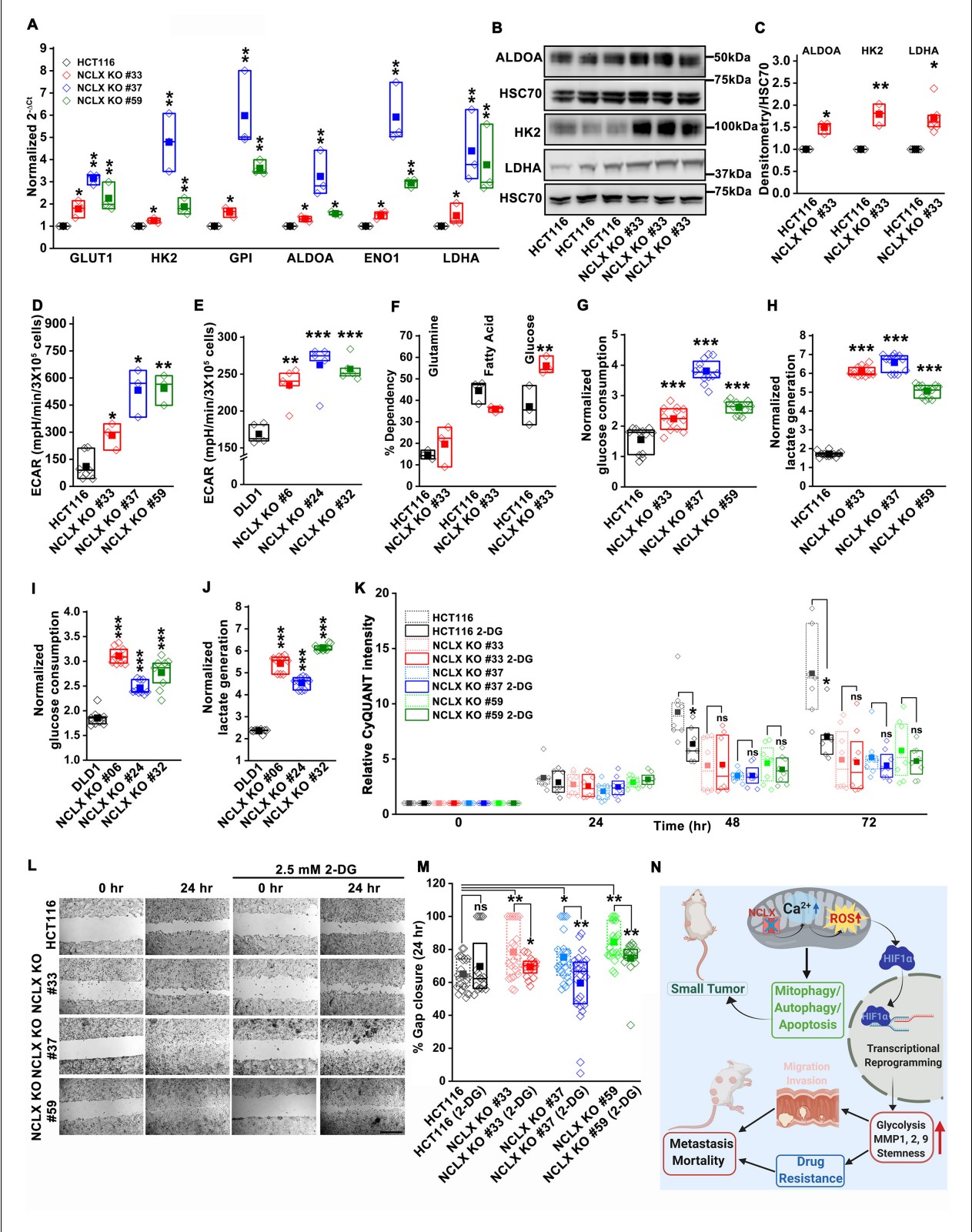

**Figure 7.** Glycolysis is critical for migration of NCLX-deficient colorectal cancer cells. (**A**) RT-qPCR analysis of glycolytic gene expression in HCT116 cells and three clones of HCT116 NCLX KO cells. RT-qPCR results are plotted as 2$^{-\Delta Ct}$ relative to tubulin and normalized to control. (**B, C**) Representative western blots probed with anti-ALDOA, anti-HK2, and anti-LDHA antibodies in HCT116 cells and HCT116 NCLX KO #33 cells. Anti-HSC70 antibody is used as a loading control (**B**) and quantification of band intensity relative to that of HSC70 (**C**). (**D, E**) Extracellular acidification rate (ECAR) of HCT116

*Figure 7 continued on next page*

*Figure 7 continued*

cells (D) and DLD1 cells (E) and their respective NCLX KO clones were measured with Seahorse using the glycolysis stress test, as described in methods. (F) The percentage of metabolite dependency of HCT116 cells and HCT116 NCLX KO #33 cells measured using the Mitochondrial Fuel Flexibility, Dependency, and Capacity test, as described in methods. (G–J) Measurement of glucose consumption and lactate generation in HCT116, clones of HCT116 NCLX KO (G, H), DLD1, and clones of DLD1 NCLX KO (I, J) cells normalized to the amount of protein. (K) Proliferation of HCT116 cells and clones of HCT116 NCLX KO cells in the presence and absence of 2.5 mM 2-DG. (L, M) In vitro migration (L), and quantification (M) of HCT116 cells and clones of HCT116 NCLX KO cells in the presence and absence of 2.5 mM 2-DG for 24 hr (scale bar, 1 mm). (N) Summary of the findings from the present study. All experiments were performed ≥three times with similar results. Statistical significance was calculated using paired t-test, except for K, M, where one-way ANOVA followed by a post-hock Tukey test was used. *p<0.05, **p<0.01, and ***p<0.001.

The online version of this article includes the following source data and figure supplement(s) for figure 7:

**Source data 1.** Source data *Figure 7*.
**Figure supplement 1.** Loss of NCLX enhances glycolysis and diminishes mitochondrial respiration.
**Figure supplement 1—source data 1.** Source Data *Figure 7—figure supplement 1*.

inhibited metastasis to the lung in xenograft experiments (*Tosatto et al., 2016*), suggesting that although mtCa$^{2+}$ overload is pro-apoptotic, it is likely beneficial for invasion and metastasis of cancer cell clones that have evaded apoptosis. Similarly, our TCGA data analysis of colorectal cancer patients showed that *SLC8B1* mRNA levels are significantly reduced in both early and late-stage tumors with a more pronounced reduction in late-stage tumors, suggesting that mtCa$^{2+}$ overload through reduced NCLX function has a critical role in CRC progression. Nevertheless, we cannot rule out possible contributions of cytosolic Ca$^{2+}$ to the phenotype of NCLX KO cells. Furthermore, future studies investigating the role of MCU in CRC progression are required to support the notion that mtCa$^{2+}$ per se is the critical driver of CRC progression.

It is well established that mtCa$^{2+}$ overload is detrimental to mitochondrial function and is a precursor of death in normal cells through the opening of the mitochondrial permeability transition pore, cytochrome C release from mitochondria and subsequent activation of the caspase family of pro-apoptotic proteases (*Giorgi et al., 2012*; *Pinton et al., 2008*). However, the exact mechanisms by which cancer cells adapt and survive under these conditions are not fully understood. Our data show that downregulation or complete loss of NCLX in CRC cells causes mtCa$^{2+}$ overload, an increase in mtROS production and mitochondria depolarization. Our transmission electron microscopy data show that loss of NCLX in CRC cells causes altered mitochondria shape and morphology with disrupted cristae and inner mitochondrial membrane structures. We also show that this coincides with increased LC3BII- and p62-dependent autophagosome formation and decreased cell-cycle-related gene expression. This phenotype is consistent with the reduced proliferation of NCLX KO and NCLX KD CRC cells, and the reduced tumor burden and tumor size in NCLX KO mice subjected to the colitis-associated CRC model. Similarly, our xenograft model yielded smaller primary tumors when HCT116 NCLX KO cells were injected in SCID mice by comparison to xenografts of control HCT116 cells. Thus, the reduction in NCLX expression likely limits proliferation and primary tumor growth. Yet, it is also apparent that clones of cancer cells survive this loss of NCLX and coopt the downregulation of NCLX to undergo metabolic reprogramming and gene expression changes that support tumor migration, invasion and metastasis, by initiating a mitochondrial Ca$^{2+}$/ROS signaling axis to drive HIF1α activation. Increased p62 is a marker for autophagosome formation but upon autophagy the levels of p62 decrease (*Liu et al., 2016*). Interestingly, our NCLX KO cells have maintained an increase in p62 levels, suggesting possible accumulation of autophagosomes. Future studies are required to determine whether increased autophagosome formation is accompanied by increased mitophagy or whether accumulation of autophagosomes in NCLX KO cells causes cytotoxicity and apoptosis (*Button et al., 2017*). We have shown that cleaved caspase three staining was increased in HCT116 NCLX KO cells, suggesting enhanced apoptosis in NCLX KO cells. However, additional experiments including more direct approaches are required to determine the effect of NCLX deletion on apoptosis of CRC cells. Herein, we have considered studying the effects of NCLX overexpression on metastasis of CRC to complement out NCLX knockout and knockdown studies. However, these studies are not feasible because overexpressed NCLX localizes not only to mitochondria but also to other organelles, making the interpretation of results from these experiments tenuous.

While it is generally thought that $mtCa^{2+}$ induces mtROS production through increased activation of TCA cycle proteins and consequential increases in ETC function, it is also evident that defective electron transport chain function results in mtROS production (*Starkov, 2008*). Our data clearly demonstrate that increased mtROS production is accompanied by mitochondrial structural perturbations, decreased OXPHOS, and mitochondrial membrane depolarization. These phenotypes of mitochondrial dysfunction are commonly associated with apoptosis initiation. However, NCLX knockout resulted in the selection of clones that initiate pro-survival and pro-metastatic adaptations. Mitochondrial depolarization and mtROS production are known initiators of mitophagy (*Ashrafi and Schwarz, 2013*; *Fan et al., 2019*; *Frank et al., 2012*; *Schofield and Schafer, 2020*; *Twig and Shirihai, 2011*). Moreover, HIF1α, which is regulated in a $mtCa^{2+}$/mtROS-dependent manner in response to NCLX loss in CRC, also contributes to mitophagy, by upregulating the pro-mitophagy proteins BNIP3 and NIX (*Bellot et al., 2009*; *Chourasia et al., 2015*; *Zhang et al., 2008*; *Zhang and Ney, 2009*).

The gene expression signatures and phenotypes observed in response to NCLX downregulation largely mirror those of the mesenchymal CRC subtype labeled as CMS4. Shared pathways include the increase in EMT, TGF- β, matrix remodeling, stemness, and decreases in Myc and cell-cycle progression (*Guinney et al., 2015*). No single mutation is solely associated with one of the CMS or CRIS subtypes, and at present, the underlying drivers of colorectal cancer molecular subtypes still remain to be fully elucidated (*Guinney et al., 2015*). However, it is interesting to note that ~ 50% of CMS4 tumors display TP53 mutations and generally lack mutations in BRAF, and that loss of NCLX is statistically linked to TP53 mutant tumors and wild type BRAF CRC tumors. We used two different CRC cell lines HCT116 and DLD1; both of these cell lines are derived from male CRC patients carrying a mutation in KRAS and PIK3CA. TCGA data analysis showed no difference in NCLX expression between CRC patients bearing the wildtype KRAS and PIK3CA and their respective mutations. Interestingly, we also saw lower basal NCLX expression in DLD1 cells that have mutated TP53 (S241F), compared to HCT116 cells, which are wildtype for TP53. Whether increased $mtCa^{2+}$ and subsequent mtROS signaling is one of the underlying drivers of CMS4 tumors remains to be determined.

A metabolic shift towards aerobic glycolysis is a hallmark of cancer progression (*Burns and Manda, 2017*; *Liberti and Locasale, 2016*). Our data demonstrated that the loss of NCLX leads to reduced oxygen consumption, increased glycolysis, and increased transcription of major glycolytic enzymes. This increased transcription of glycolytic enzymes is commonly controlled by HIF1α, which is upregulated in response to increased mtROS resulting from loss of NCLX and subsequent increase in $mtCa^{2+}$. Furthermore, we show that inhibiting glycolysis of NCLX KO CRC cells partially normalizes the increased migration of these cells, consistent with previous findings that showed a critical role of glycolysis in cancer metastasis (*Bu et al., 2018*; *Gillies et al., 2008*; *Han et al., 2013*). Although the increase in HIF1α protein levels was shown to be critical for the migration and metastasis of cancer cells (*Lehmann et al., 2017*; *Masoud and Li, 2015*; *Semenza, 2003*; *Wigerup et al., 2016*), the mechanisms by which $mtCa^{2+}$ regulates HIF1α protein levels are not clear. Here, we provide evidence that reduced NCLX function, causing reduced $mtCa^{2+}$ extrusion and resulting in $mtCa^{2+}$ overload, enhances HIF1α levels through an increase in mtROS. Our xenograft model shows that despite the reduced tumor burden and tumor size in SCID mice injected with HCT116 NCLX KO cells compared to SCID mice injected with control HCT116 cells, the survival of the former mice was significantly reduced compared to the latter. This result can be explained by the enhanced metastasis to the liver and colon in SCID mice injected with HCT116 NCLX KO cells. Indeed, we were able to show that the increased invasive properties of NCLX KO CRC cells were associated with increased MMP1, MMP2, and MMP9 activities, which are known to be regulated by HIF1α (*Ben-Yosef et al., 2002*; *Choudhry and Harris, 2018*; *Muñoz-Nájar et al., 2006*; *Shin et al., 2015*; *Tsai et al., 2016*). We showed that NCLX KO CRC cells are chemoresistant to treatment by 5-FU, with both proliferation and migration of NCLX KO CRC cells not significantly altered by 5-FU treatment. It is reasonable to suspect that the decrease in proliferation contributes to the chemoresistance phenotype of cells lacking NCLX. In previous work, we demonstrated that decreased $mtCa^{2+}$ extrusion in HEK293 cells leads to a decrease in cytosolic $Ca^{2+}$ and reduced store-operated $Ca^{2+}$ entry (SOCE) mediated by ORAI1 channels (*Ben-Kasus Nissim et al., 2017*). Similarly, we see a decrease in cytosolic $Ca^{2+}$ and SOCE when NCLX is knocked out in CRC cell lines. A consequence of reducing SOCE and cytosolic $Ca^{2+}$ is decreased activation of the $Ca^{2+}$ sensitive transcription factor NFAT (*Trebak and Kinet, 2019*). NFAT is a known stimulator of MYC activation (*Buchholz et al., 2006*; *Mognol et al., 2012*;

*Singh et al., 2010*), suggesting that reduced NFAT signaling as a consequence of NCLX downregulation might be contributing to the decrease in MYC pathway activation, cell-cycle progression, and proliferation.

The EMT phenotype and decreases in proliferation are hallmarks of cancer stem cells and the presence of cancer stem cells within tumors is one of the major causes of chemotherapy resistance (*Izumiya et al., 2012*; *Munro et al., 2018*; *Ohata et al., 2019*; *Zhao, 2016*). Studies have shown that CSCs are enriched on treatment with 5-FU through different molecular mechanisms (*Abdullah and Chow, 2013*; *Lu et al., 2015*; *Touil et al., 2014*). Recent work showed that increased chemoresistance of breast cancer was mediated by HIF1α-mediated glutathione biosynthesis and glutathione-mediated enrichment with cancer stem cells (*Lu et al., 2015*). Chemotherapy in breast cancer induces HIF1α-dependent glutathione biosynthesis by enhancing the expression of the cystine transporter xCT (SLC7A11) and the regulatory subunit of glutamate-cysteine ligase (GCLM) (*Lu et al., 2015*). The glutathione thus generated inhibits the MEK/ERK pathway through copper chelation resulting in enhanced expression of the stem cell markers NANOG, Oct4 and Sox2. We observed enhanced expression of SLC7A11 and GCLM in NCLX KO clones from both HCT116 and DLD1 cells. In addition to contributing to stem cell regulation, an increase in enzymes responsible for glutathione synthesis potentially provides additional survival advantages to NCLX knockout cells, by enhancing the ROS scavenging ability of CRC cells and preventing any lethal build-up of ROS as a consequence of mtCa$^{2+}$ elevation. Throughout this study, HIF1α emerged as major regulator of metastasis, chemoresistance and stemness in NCLX KO CRC cells. HIF1α might represent a novel marker or target of anti-CRC drugs. One could speculate that the concomitant inhibition of HIF1α and chemotherapy might be more effective against CRC cancer.

In summary, we have identified a novel mitochondrial Ca$^{2+}$/ROS signaling axis in colorectal cancer that is initiated in response to decreased NCLX expression, as commonly observed in CRC patient tumors. In one hand, reduced mtCa$^{2+}$ extrusion causes mitochondrial depolarization, mitochondrial dysfunction, reduced cell-cycle-related gene expression, increased autophagosome formation, and reduced proliferation, leading to smaller tumors both in the colitis-associated CRC model and the xenograft CRC model. On the other hand, mtCa$^{2+}$ overload induced by NCLX loss enhances mtROS, which in turn enhances CRC metastasis through HIF1α-dependent increases in glycolysis, chemoresistance and pro-metastatic gene expression signatures. This contributes to increased metastatic burden and enhanced lethality of SCID mice injected with NCLX KO CRC cells (summarized in *Figure 7N*). These changes are reminiscent of the highly metastatic mesenchymal CMS4 CRC subtype, and our work suggests that restoring mtCa$^{2+}$ homeostasis in CRC tumors might be beneficial in limiting or preventing CRC progression and metastasis.

## Materials and methods

### Reagents and resources
See article appendix for key resources table.

### Human tissue
The Pennsylvania State University College of Medicine institutional review board approved this study (IRB Protocol number HY98-057EP-A). Prior to surgery, patients are consented to have resected tissues collected and banked into the Penn State Hershey Colorectal Disease Biobank. As outlined in the consent form and IRB Protocol, de-identified tissues are made available to study on a per-request basis after such requests are considered by an internal review panel and deemed to be collaborative in nature. The CRD biobank has been in operation since 1998 and substantial efforts are in place to protect patient information including storage of data behind firewalls and limited access to electronic medical records. Methods regarding tissue procurement, mRNA isolation and processing were previously described (*Schieffer et al., 2017*).

### Mice
A breeding pair of global NCLX KO and wildtype C57BL/6 mice were obtained from the Jackson laboratory. NOD/SCID (NOD.CB17-*Prkdc$^{scid}$*/J) were also obtained from the Jackson laboratory. All mice were maintained in 12:12 hr light-dark cycle (22°C to 25°C) in a pathogen-free barrier facility at

The Pennsylvania State University animal facility. The mice were sacrificed (age at the time of sacrifice is mentioned in the figure legends) at the end of the experiment, and organs were collected for histopathology, protein, and mRNA analysis. All experiments with mice were approved and conducted in accordance with The Pennsylvania State Institutional Animal Care and Research Advisory Committee.

## Cell culture and drug treatment

To investigate the role of NCLX in human CRC cells, we chose two widely used CRC cell lines HCT116 and DLD1 derived from the colon of male CRC patients with colorectal carcinoma and colorectal adenocarcinoma, respectively (*Ahmed et al., 2013*). HCT116 and HT29 cells were cultured in McCoy's 5A media supplemented with 10% FBS and 1X antibiotic-antimycotic agents. DLD1 cells were cultured in RPMI-1640 media supplemented with 10% FBS and 1X antibiotic-antimycotic agents. All cells were kept at 37°C in a 5% $CO_2$ incubator. The cells were transfected with siRNA or plasmids using the Lipofectamine 2000 reagent, according to manufacturer protocol (Invitrogen). In experiments with 2-DG and 5-FU, cells were cultured in complete growth media McCoy's or RPMI media with 10% FBS and 1X antibiotic-antimycotic agents and treated with different concentrations of 2-DG or 5-FU for the specified time. Our cell lines were purchased from ATCC immediately before use and the identities of the cell lines were further confirmed by RNA sequencing. Further, we routinely perform mycoplasma tests on all our cell lines, including the cell lines used herein. These mycoplasma tests were negative.

## Plasmids, siRNAs, shRNAs, CRISPR/Cas9, and lentiviral infection

The cells were seeded at 80–90% confluency at the time of transfection. The plasmids or siRNAs and Lipofectamine 2000 were diluted in Opti-MEM and mixed. The mixture was incubated for 5–10 min at room temperature and then added to the cells cultured in media with 10% serum without antibiotic-antimycotic agents. The medium of transfected cells was changed to normal media with 10% FBS and 1X antibiotic-antimycotic agents after 4–6 hr. The expression of plasmid and siRNA was confirmed after 24 hr and 72 hr of transfection, respectively. To generate stable NCLX knockdown cells, shScramble, shNCLX #2, and shNCLX #3 cloned in pLKO. The lentiviral construct was packaged into HEK293FT using the ViraPower kit (Invitrogen) and Lipofectamine 2000 (Thermofisher Scientific). HCT116 cells were infected with respective lentivirus for 72 hr and selected with puromycin (1.5 µg/ ml) for 72 hr. The knockdown of mRNA was confirmed through RT-qPCR. Stable knockdown cells were used within two weeks, then discarded. We generated several clones of NCLX knockout (NCLX KO) in both HCT116 and DLD1 cells using the CRISPR/Cas9 system. To alleviate potential off-target effects of the CRISPR/Cas9 system, we generated several independent clones obtained with three independent guide RNAs (gRNAs; see methods) (*Figure 3—figure supplement 1A–H*). Genome sequencing and PCR on genomic DNA confirmed NCLX KO. For the case of HCT116 cells, which we generated first, NCLX KO #33 was generated using a guide RNA (g2) which resulted in a single cut at nucleotide 150 in exon one causing a frameshift mutation and introduction of a stop codon at predicted position 181-183 bp and 184-186 bp in the NCLX open reading frame (*Figure 3—figure supplement 1A*). NCLX KO #37 and #59 were generated using two guide RNAs (g$_1$ and g$_2$), which resulted in a double-strand cut and introduction of a stop codon in the NCLX open reading frame (*Figure 3—figure supplement 1B,C*; method table). For the NCLX KO clones of DLD1 generated later, we employed an advanced strategy with each knockout condition using simultaneously two guide RNAs flanking a region starting from exon three to the end of the *SLC8B1* gene (*Figure 3— figure supplement 1E*) to completely excise most of NCLX open reading frame. PCR on genomic DNA (*Figure 3—figure supplement 1G*) confirmed NCLX knockout in DLD1 clones. We used RT-qPCR to document the absence of mRNA in NCLX KO clones of HCT116 (*Figure 3—figure supplement 1D*) and DLD1 cells (*Figure 3—figure supplement 1H*). The positions of the primers used for qPCR are shown in *Figure 3—figure supplement 1A* and *Figure 3—figure supplement 1E* for HCT116 and DLD1 cells, respectively, and labeled as 'qNCLX'.

## Transcriptome sequencing (RNA-seq) and analysis

The mRNA was isolated (minimum 3 µg) using the RNeasy Mini Kit. A fragmentation buffer (Ambion) was used to short fragment the mRNA. These short-fragmented mRNA were used as a template to

synthesize double-stranded cDNA. The cDNA was end-repaired and ligated to Illumina adapters. Further, to generate the library, these cDNAs were size selected (~250 bp) on an agarose gel, and PCR amplified. To sequence the cDNA, the prepared cDNA library was fed to the Illumina HiSeq 2500 sequencing platform (Berry Genomics). The expression level of each transcript of a gene was quantified using read counts with HTseq. After removing gene identifiers with zero read counts in each sample, the biomaRt R package (*Durinck et al., 2009*) was used to convert Ensemble gene identifiers to gene symbols using the January 2019 archive (http://jan2019.archive.ensembl.org). Identifiers corresponding to multiple gene symbols were removed. The edgeR R package (*McCarthy et al., 2012*; *Robinson et al., 2010*) was used to identify lowly expressed genes based on thresholds defined using counts per million reads mapped. This yielded expression data for a total of 10,179 genes.

Exploratory analyses were performed after utilizing the variance stabilizing transformation (VST) function in the DESeq2 R package (*Love et al., 2014*) to apply VST to the read count data. The results strongly suggested the presence of a batch effect. After applying the voom transformation (*Law et al., 2014*) to the read counts, the limma R package (*Ritchie et al., 2015*) was used to identify differentially expressed genes based on a q-value threshold of 0.05. The limma design matrix included batch information as a factor.

## Gene set enrichment analysis (GSEA)

The gene set enrichment analysis (GSEA) software (*Subramanian et al., 2005*) was used to perform pathway analyses based on the limma output. Briefly, genes in the limma output were ordered according to the *t* test statistic. Then a GSEA pre-ranked analysis was performed using the Hallmark Gene Sets in the Molecular Signatures Database (http://software.broadinstitute.org/gsea/msigdb/index.jsp).

## TCGA analysis

RNA-sequencing (RNA-seq)-based gene expression data for both tumor and normal samples, as well as clinical data for the combined TCGA COADREAD cohort were downloaded from the Broad Institute's Firehose GDAC (https://gdac.broadinstitute.org/). Gene expression was quantified as log2 (normalized RSEM + 1). Somatic gene mutation data was obtained from *Ellrott et al., 2018* after restricting to samples in the RNA-seq cohort. Wilcoxon rank-sum tests and Kruskal-Wallis tests were used to compare expression values of *SLC8B1* across groups defined by the clinical and mutation data.

## Reverse transcription-quantitative polymerase chain reaction (RT-qPCR)

Human colorectal cancer tissues utilized for mRNA extraction were obtained from the Penn State Hershey medical center with IRB approval. Briefly, mRNA were isolated from either tissues or cells. The cells were washed with cold PBS, and RNA was isolated using the RNeasy Mini Kit. The quality and quantity of RNA were analyzed using NanoDrop2000 (Thermo Scientific, Wilmington, DE, USA). High-Capacity cDNA Reverse Transcription Kit (Applied Biosystems, Foster City, CA) was used to make cDNA. The cDNA was further used for quantitative real-time PCR using the SYBR select master mix (Thermo Scientific) according to the manufacturer protocol. The reactions were run in technical duplicates using the Quant Studio three system (Applied Biosystems). The expression levels of genes were normalized to at least two housekeeping genes, GAPDH or Tubulin or NONO (specified in the figure legends) using the $2^{-\Delta Ct}$ method.

## Western blot

The cells were cultured with 80–90% confluency and were washed with chilled PBS and lysed in 100 µl RIPA buffer (Sigma) containing Halt protease and phosphatase inhibitor. 30–50 µg of the protein was loaded on a 4–12% gel (NuPAGE Bis-Tris precast gels, Life Technologies) and transferred to a polyvinylidene difluoride membrane. The membrane was incubated in Odyssey blocking buffer for 1 hr at room temperature for blocking and was incubated overnight at 4°C in primary antibody diluted in Odyssey blocking buffer. The primary antibodies were used are mentioned in the key resource table, and their dilutions are as follows: anti-HIF1α (1:500), all other antibodies were used at 1:1000 dilution. The next day, the membrane was washed with 0.1% TBS-T for 5 min (three times) at room

temperature, followed by incubation in IRDye for 2 hr at room temperature. The IRDyes used are mentioned in the key resource table, and they were diluted in Odyssey blocking buffer (IRDye 680RD Goat anti-Mouse at 1:10,000 dilution and IRDye 800CW Donkey anti-Rabbit at 1:5000). The membrane was washed with 0.1% TBS-T for 5 min (three times) at room temperature. Then the blot was imaged in Odyssey CLx Imaging System (LI-COR, NE, USA). Densiometric analysis of the bands on the membrane was performed using ImageJ software.

## Immunofluorescence microscopy

The cells were cultured in six well plates on glass coverslip at 60–70% confluency. The next day, the cells were washed with cold PBS and fixed with chilled 100% Methanol for 30 min. The cells were washed with chilled PBS (3X for 5 min). The cells were incubated in an anti-cleaved caspase-3 antibody (1:250). Secondary Alexa Fluor 594 (1:5000) was used to stain the cells.

For live-cell imaging and mitochondrial staining, the cells were stained with MitoTracker Green FM (50 nM in complete media for 30 min at 37°C), TMRM (100 nM in complete media for 30 min at 37°C) or MitoSOX Red (2.5 µM in HBSS for 30 min at 37°C) and Hoechst (1 µM in HBSS for 5 min). The images were acquired using the Leica SP8 confocal microscope. The fluorescence intensity was quantified using LAS X (Leica Microsystems) software.

## Mitochondrial $Ca^{2+}$ measurements

To measure mitochondrial $Ca^{2+}$ using Rhod-2 AM (543 nm/580–650 nm) dye, the cells were cultured at 50–60% confluency. The cells were washed with media without FBS and antibiotic-antimycotic agents. Then the cells were incubated in media containing 1 µM Rhod-2 AM (without FBS and antibiotic-antimycotic agents) at 37°C for 30 min. The cells were washed with media and were loaded with MitoTracker Green FM (50 nM in media for 30 min at 37°C) for photobleaching and focus corrections. After loading with MitoTracker Green, cells were washed and kept in HBSS (HEPES-buffered saline solution (140 mM NaCl, 1.13 mM $MgCl_2$, 4.7 mM KCl, 2 mM $CaCl_2$, 10 mM D-glucose, and 10 mM HEPES, adjusted to pH 7.4 with NaOH)) containing 5 mM $CaCl_2$ for imaging. The cells were stimulated with 300 µM ATP in HBSS containing 5 mM $CaCl_2$. The time-lapse images were acquired using the Leica TCS SP8 confocal microscope equipped with a 63X oil objective. The images were acquired every 5 s intervals for 10–15 min. The fluorescence intensity was quantified using LAS X (Leica Microsystems) software.

## Intracellular $Ca^{2+}$ measurements

Intracellular $Ca^{2+}$ imaging was performed described previously (*Emrich et al., 2019*; *Zhang et al., 2019*). Briefly, the cells were cultured on glass coverslips with 50–60% confluency. Next day the cells were washed with fresh media, and the glass coverslip with cells were mounted on a Teflon chamber. Further, the cells were incubated at 37°C for 45 min in culture media containing 4 µM Fura-2/acetoxymethyl ester (Molecular Probes, Eugene, OR, USA). Then the cells were washed and kept in HEPES-buffered saline solution (140 mM NaCl, 1.13 mM $MgCl_2$, 4.7 mM KCl, 2 mM $CaCl_2$, 10 mM D-glucose, and 10 mM HEPES, adjusted to pH 7.4 with NaOH) for at least 10 min before $Ca^{2+}$ measurements were made. Time-lapse imaging was performed, and the change in fluorescence of Fura-2 was recorded and analyzed with a digital fluorescence imaging system (InCyt Im2; Intracellular Imaging Inc, Cincinnati, OH, USA). An ROI was drawn around each cell, and 340/380 ratio of each cell was calculated.

## Transmission electron microscopy (TEM)

The HCT116, DLD1, NCLX KO HCT116, and NCLX KO DLD1 cells were cultured at 80–90% confluency and fixed with 1% glutaraldehyde in 0.1 M sodium phosphate buffer, pH 7.3. After fixation, the cells were washed with 100 mM Tris (pH 7.2) and 160 mM sucrose for 30 min. The cells were washed again twice with phosphate buffer (150 mM NaCl, 5 mM KCl, 10 mM $Na_3PO_4$, pH 7.3) for 30 min, followed by treatment with 1% $OsO_4$ in 140 mM $Na_3PO_4$ (pH 7.3) for 1 hr. The cells were washed twice with water and stained with saturated uranyl acetate for 1 hr, dehydrated in ethanol, and embedded in Epon (Electron Microscopy Sciences, Hatfield, PA). Roughly 60 nm sections were cut and stained with uranyl acetate and lead nitrate. Further, the stained grids were analyzed using a Philips CM-12 electron microscope (FEI; Eindhoven, The Netherlands) and photographed with a

Gatan Erlangshen ES1000W digital camera (Model 785, 4 k 3 2.7 k; Gatan, Pleasanton, CA). Morphometric analysis of mitochondrial was analyzed by double-blinded independent observers in at least 10 different micrographs per condition. The mitochondria without intact inner mitochondrial membrane and cristae were classified as damaged, and their number was counted in each cell and divided by the total number of mitochondria in that cell to calculate % damaged mitochondria. The mitochondrial area was measured by manually tracing the outer mitochondrial membrane and using the measure function of NIH ImageJ software. Cristae per mitochondria were calculated by counting the number of intact cristae in each mitochondrion.

## Mitochondrial membrane potential and mitochondrial ROS measurements

Cells were cultured in 6-well plates at 50–60% confluency. The next day, $1 \times 10^6$ cells were harvested and loaded with Tetramethyl rhodamine (TMRM) dye. The cells were stained with 100 nM TMRM dye in complete growth media and kept at 37°C in 5% $CO_2$ for 20–30 min. CCCP (50 µM) was used as a positive control. To measure mitochondrial ROS, $1 \times 10^6$ cells were stained with Mito-SOX (2.5 µM in HBSS at 37°C for 30 min). Antimycin (50 µM) was used as a positive control. The intensity of staining was measured on an LSRII flow cytometer using FACSDiva software (BD Biosciences) and analyzed with FlowJo software (Tree Star).

## Cell proliferation assays

The cells were harvested, and 3000–5000 cells were plated in each well of 96 well plates. The cells were kept at 37°C in 5% $CO_2$ for 4 hr to allow the cells to adhere to the plate. The CyQUANT-NF dye was diluted in HBSS buffer, and 100 µl of the mixture was added in each well. The plate was kept at 37°C in 5% $CO_2$ for 1 hr. The fluorescence intensity (~485 nm/~530 nm) was measured using FlexStation 3 Multimode Plate Reader (VWR). To analyze the effect of 2-DG and 5-FU on proliferation, different doses of the drugs were added in the culture media and cells were grown for the indicated amount of time. The fluorescence intensity of the dye was recorded after the drug treatment. The normalized intensity was calculated using the following formula:

Normalized intensity = $(I_t-I_b)/(I_0-I_b)$
$I_t$ = intensity at the given time
$I_b$ = background intensity
$I_0$ = intensity at time zero

## Zymography

Briefly, the cells were harvested and cultured in 6-well plates. Once the culture becomes 80–90% confluent, 20 µl of the media was taken and mixed with Tris-Glycine-SDS sample buffer and loaded in a Novex Zymogram gel and Tris-glycine SDS running buffer was used to run the gel. After electrophoresis, the gel was kept in Novex Zymogram renaturing buffer for 30 min and transferred to Novex Zymogram Developing buffer and kept at 37°C overnight. The next day the gel was stained with SimplyBlue Safe Stain and imaged using FluorChem M imager. The images were analyzed using ImageJ.

## In vitro migration and invasion assays

Migration was measured using two different methods a transwell migration assay and wound healing assay. For the transwell migration assay using FluoroBlok, the cells were serum-starved for 24 hr, and 1,000 cells were plated on the chamber in 200 µl media without FBS. The cells were stimulated to migrate towards preconditioned media with 10% FBS. After 8 hr of migration, the upper part of the membrane was cleaned, and the lower part was stained with DAPI, and the intensity of DAPI was measured using FlexStation 3 Multimode Plate Reader (VWR).

For the wound healing (gap closure) assay, cells were serum-starved for 24 hr to synchronize the cell cycle. Then 50,000 cells were plated in ibidi-silicone insert with a defined cell-free gap. After 24 hr, the inserts were lifted, and the gap between two migrating fronts was measured at 0 hr, 12 hr, and 24 hr using Zeiss inverted fluorescence microscope equipped with 5X air objective. The area between the two migrating fronts was measured using ImageJ (NIH, Bethesda). To analyze the effect of 2-DG and 5-FU on migration, a specific dose of the drugs was added in the culture media at the

time of removal of the silicon insert, and cells were grown for the indicated amounts of time. Fresh media with the drug was added every 12 hr. The following formula was used to calculate % gap closure:

% Gap closure = $(A_0-A_t/A_0)/100$
$A_0$ = area of gap measured immediately after lifting the insert
$A_t$ = area of gap measured after the indicated time of migration

Cell invasion was measured using the BioCoat Tumor Invasion Plate with 8 µm pore size inserts, which were coated with growth factor-reduced Matrigel. The cells were serum-starved for 24 hr, and a total of 5,000 cells were plated on the chamber in 200 µl media without FBS. The cells were stimulated to invade towards preconditioned media with 10% FBS. After 24 hr invasion, the upper part of the membrane was cleaned, and the lower part was stained with DAPI, and the intensity of DAPI was measured using FlexStation 3 Multimode Plate Reader (VWR). The percentage of invasion was calculated using the following formula-

$$\%\text{invasion} = (\text{Fluorescence of invading cells} - \text{Fluorescence of blank chamber})/$$
$$(\text{Fluorescence of migrating cells} - \text{Fluorescence of blank chamber})\text{X}100$$

## ECAR and oxygen consumption rate (OCR)

The ECAR, OCR, and % metabolite dependency was measured using the Seahorse XFp Extracellular Flux Analyzer (Seahorse Bioscience). All experiments were performed according to the manufacturer's protocol. Seahorse XFp Glycolysis Stress Test Kit, Seahorse XFp Cell Mito Stress Test Kit, and Seahorse XFp Mito Fuel Flex Test Kit (Agilent Technologies) were used to measure ECAR, OCR, and % dependency respectively. Briefly, cells were harvested, and 30,000 cells were plated per well and cultured in a Seahorse XFp cell culture microplate with complete growth media for 10–12 hr. The next day the cells were washed with Seahorse XF DMEM media (pH 7.4). The plate with cultured cells was placed in the Seahorse XFp Extracellular Flux Analyzer, and baseline measurements were recorded. For ECAR measurements, oligomycin, and 2-DG were sequentially injected into each well at the indicated time points. Similarly, for OCR measurements, oligomycin, FCCP (p-trifluoromethoxy carbonyl cyanide phenylhydrazone), and antimycin A (Rote/AA) were sequentially injected. For % dependency measurements, UK5099, BPTES, and Etomoxir were sequentially injected. The results obtained for ECAR, OCR, and Mito Fuel Flex Test were normalized to cell number. Data were analyzed using Seahorse XFp Wave software. The results for OCR are reported in pmols/minute, ECAR in mpH/minute, and Mito Fuel Flex Test in % dependency.

## Glucose and lactate measurements

To measure glucose and lactate in media, 100,000 cells were plated in each well of 6-well plates with 2 ml of complete growth media. After 24 hr, 400 µl of the media were taken from each well of the 6-well plates and divided into four wells in a 96-well plate (quadruplicate for each NCLX KO clone and respective control). Then the plate was kept in YSI 7100 multichannel biochemistry analyzer (YSI Life Sciences) to measure glucose and lactate levels in the media. Fresh media was used to measure the basal level of glucose and lactate. After measurements were completed, cells were harvested, and the protein content was measured using the BCA assay. This experiment was performed at least three times independently. The following formulas were used to calculate glucose consumption and lactate generation:

Glucose consumption = (Glucose in fresh media - Glucose in the media with cells)/protein content
Lactate generation = (Lactate in the media with cells - Lactate in fresh media)/protein content

## Mouse xenograft experiments

Mice were anesthetized using isoflurane, and after proper sterile preparation of the abdomen, a small (4–8 mm) incision was made by sharp sterile blade over the left upper quadrant of the abdomen. The peritoneal cavity was carefully exposed, and the spleen was located. Further, the spleen was carefully exteriorized on the sterile field around the incision area, and 500,000 cells suspended in 50 µl McCoy's (10 or 20% FBS) were then injected into the spleen via a 1 ml insulin syringe. A small bleb was observed while injecting the cells in the spleen, confirming that the cells were

successfully injected. The spleen was carefully placed back into the peritoneal cavity. Both the peritoneal cavity and skin were closed with sterile absorbable suture. The mice injected with luciferin (100 µg/ml) and imaged in the IVIS system every week till the time of sacrifice.

## Colitis-associated colon cancer model and histology

6–8 week-old mice were given a single azoxymethane (AOM) injection via IP. One week after AOM injection, the mice were given three cycles of dextran sodium sulfate (DSS), which was supplemented in drinking water (1.5 % m/v, MW 36–50 kDa, MP Biochemicals) ad libitum for five days. The DSS cycles were intermixed with two weeks of normal autoclaved drinking water. Mice were sacrificed, and the colons were collected ten weeks after the last DSS cycle. The collected colons were flushed with PBS to clear feces and photographed. Before dissection, tumors were scored according to their number and size. PBS-rinsed colons were either snap-frozen for further analysis or fixed in 10% neutral buffered formalin and sectioned for histological analysis.

## Quantification and statistical analysis

Data are represented as mean ± SEM and analyzed using Origin pro 2019b (Origin lab). In the box plot, the box represents the $25^{th}$ to $75^{th}$ interquartile range, midline in the box represents the median, and the solid square box represents mean data points. To test single variables between two groups, paired t-test or Kruskal-Wallis ANOVA was performed (specified in each figure legend). One-way ANOVA followed by post-hoc Tukey's test was used for comparison between multiple conditions and the control group. The $p$-value$<0.05$ was considered to be significant and is presented as $*p<0.05$, $**p<0.01$, or $***p<0.001$.

## Acknowledgements

The authors acknowledge Dr. Han Chen from The Pennsylvania State University College of Medicine EM facility for assistance with TEM imaging. Our study was supported by the National Heart, Lung, and Blood Institute (R01-HL123364, R01-HL097111, and R35-HL150778 to MT), National Institute on Aging (R21-AG050072 to MT), and American Heart Association Postdoctoral Fellowship (9POST34380606) to TP GSY and WAK are supported by the Peter and Marshia Carlino Fund for IBD research. We also acknowledge, Flow Cytometry and Cell Sorting, Imaging, Informatics and Data Analysis Core facility (The Pennsylvania State University, College of Medicine).

## Additional information

### Funding

| Funder | Grant reference number | Author |
| --- | --- | --- |
| National Heart, Lung, and Blood Institute | R35-HL150778 | Mohamed Trebak |
| American Heart Association | 9POST34380606 | Trayambak Pathak |
| National Heart, Lung, and Blood Institute | F30 HL147489 | Martin T Johnson |
| National Heart, Lung, and Blood Institute | R01-HL123364 | Mohamed Trebak |
| National Heart, Lung, and Blood Institute | R01-HL097111 | Mohamed Trebak |
| National Institute on Aging | R21-AG050072 | Mohamed Trebak |
| Peter and Marshia Carlino Fund | | Gregory S Yochum Walter A Koltun |

The funders had no role in study design, data collection and interpretation, or the decision to submit the work for publication.

## Author contributions
Trayambak Pathak, Conceptualization, Data curation, Formal analysis, Investigation, Visualization, Methodology, Writing - original draft, Writing - review and editing; Maxime Gueguinou, Conceptualization, Investigation; Vonn Walter, Resources, Software, Formal analysis; Celine Delierneux, Martin T Johnson, Xuexin Zhang, Ping Xin, Ryan E Yoast, Scott M Emrich, Investigation; Gregory S Yochum, Israel Sekler, Resources, Writing - review and editing; Walter A Koltun, Donald L Gill, Resources; Nadine Hempel, Conceptualization, Resources, Supervision, Methodology, Writing - original draft, Writing - review and editing; Mohamed Trebak, Conceptualization, Supervision, Funding acquisition, Methodology, Writing - original draft, Project administration, Writing - review and editing

## Author ORCIDs
Vonn Walter  http://orcid.org/0000-0001-6114-6714
Israel Sekler  http://orcid.org/0000-0002-7550-1550
Mohamed Trebak  https://orcid.org/0000-0001-6759-864X

## Ethics
Human subjects: The Pennsylvania State University College of Medicine institutional review board approved this study. Approval under IRB Protocol number HY98-057EP-A. Prior to surgery, patients are consented to have resected tissues collected and banked into the Penn State Hershey Colorectal Disease Biobank. As outlined in the consent form and IRB protocol.

Animal experimentation: This study was performed in strict accordance with the recommendations in the Guide for the Care and Use of Laboratory Animals of the National Institutes of Health. All of the animals were handled according to approved institutional animal care and use committee (IACUC) protocol # 47350/Trebak, which was approved by the IACUC at the Penn State University college of medicine. Every effort was made to minimize animal suffering.

## Decision letter and Author response
Decision letter https://doi.org/10.7554/eLife.59686.sa1
Author response https://doi.org/10.7554/eLife.59686.sa2

# Additional files
## Supplementary files
• Supplementary file 1. Genomic sequencing of NCLX KO clones of HCT116 and DLD1 cells. This file shows the genome sequencing of NCLX KO clones #33, #37, and #59 of HCT116 and #06, #24, and #32 of DLD1 cells. For genomic sequencing, PCR was performed to amplify specific parts of the NCLX gene using primers listed in Appendix 1_key resources Table 1. The primers used for cloning genomic DNA from HCT116 cells were PX75 NCLX test F and PX76 NCLX test R. The primers used for cloning genomic DNA from DLD1 cells were NCLX_4 and NCLX_5. PCR products were cloned using StrataClone Blunt PCR Cloning Kit and sent to Genewiz for sanger sequencing. The genome sequencing data for the HCT116 NCLX KO clones confirm the introduction of STOP codons in coding sequences of HCT116 NCLX KO #33 at predicted positions 181-183 bp and 184-186 bp, in HCT116 NCLX KO #37 at 16-18 bp and 46-48 bp, and in HCT116 NCLX KO #59 at 31-33 bp, and 52-54 bp. Furthermore, the genomic sequencing of NCLX KO clones of DLD1 cells showed that the middle ~32 kb portion of the NCLX gene was deleted in all clones. Therefore, providing decisive evidence that these cells are indeed NCLX KO.

• Source data 1. Source data RNAseq_HCT116_HCT116 NCLX KO.

• Transparent reporting form

## Data availability
No large data sets have been generated from the current study. All data generated or analysed during this study are included in the manuscript and supporting files. Source data files for all figures and figure supplements have been provided in Source data 1.

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

# Appendix 1

**Appendix 1—key resources table**

| Reagent type (species) or resource | Designation | Source or reference | Identifiers | Additional information |
|---|---|---|---|---|
| Gene (*Homo sapiens*) | SLC8B1/NCLX | GenBank | Gene ID: 80024 | |
| Gene *Mus musculus* | Slc8b1/NCLX | GenBank | Gene ID: 170756 | |
| Genetic reagent *Mus musculus* | NOD.CB17-Prkdcscid/J | Jackson Laboratory | Stock No: 010636, RRID:IMSR_JAX:010636 | Male |
| Cell line (*Homo-sapiens*) | HCT116 (colon, epithelial) | ATCC | ATCC# CCL-247 | Male |
| Cell line (*Homo-sapiens*) | HT29 (colon, epithelial) | ATCC | ATCC# HTB-38 | Female |
| Cell line (*Homo-sapiens*) | DLD1 (colon, epithelial) | ATCC | ATCC# CCL-221 | Male |
| Cell line (*Homo-sapiens*) | HCT116 NCLX KO (colon, epithelial) | This paper | | NCLX Knockout clones of HCT116 cells were generated by the Trebak lab using CRISPR/Cas9 and are available upon request |
| Cell line (*Homo-sapiens*) | DLD1 NCLX KO (colon, epithelial) | This paper | | NCLX Knockout clones of DLD1 cells were generated in the Trebak lab using CRISPR/Cas9 and are available upon request |
| Cell line (*Homo-sapiens*) | HCT116 shNCLX KO (colon, epithelial) | This paper | | HCT116 cells with stable shRNA-mediated knockdown of NCLX (shNCLX) were generated by the Trebak Lab using shRNA sequences (listed below in this table) cloned in the lentiviral vector pLKO. These plasmids are available upon request |
| Cell line (*Homo-sapiens*) | DLD1 shNCLX KO (colon, epithelial) | This paper | | DLD1 cells with stable shRNA-mediated knockdown of NCLX (shNCLX) were generated by the Trebak Lab using shRNA sequences (listed below in this table) cloned in the lentiviral vector pLKO. These plasmids are available upon request |
| Antibody | anti-Hif1$\alpha$ (Rabbit polyclonal) | Cell Signaling Technology | Cat# 14179s, RRID:AB_2622225 | WB (1:500) |
| Antibody | anti- ALDOA (Rabbit polyclonal) | Cell Signaling Technology | Cat# 8060S, RRID:AB_2797635 | WB (1:1000) |
| Antibody | anti- HK2 (Rabbit polyclonal) | Cell Signaling Technology | Cat# 2867S, RRID:AB_2232946 | WB (1:1000) |

*Continued on next page*

*Appendix 1—key resources table continued*

| Reagent type (species) or resource | Designation | Source or reference | Identifiers | Additional information |
|---|---|---|---|---|
| Antibody | anti- LDHA (Rabbit polyclonal) | Cell Signaling Technology | Cat# 2012S, RRID:AB_2137173 | WB (1:1000) |
| Antibody | anti- MMP1 (Rabbit polyclonal) | Abcam | Cat# ab38929, RRID:AB_776395 | WB (1:1000) |
| Antibody | anti- MMP2 (Mouse monoclonal) | Santa Cruz Biotechnology | Cat# sc-13594, RRID:AB_627956 | WB (1:500) |
| Antibody | anti- MMP9 (Rabbit polyclonal) | Abcam | Cat# ab73734, RRID:AB_1860201 | WB (1:1000) |
| Antibody | anti- LC3B (Rabbit polyclonal) | Abcam | Cat# ab51520, RRID:AB_881429 | WB (1:1000) |
| Antibody | anti- OXPHOS (Mouse monoclonal) | Abcam | Cat# ab110413, RRID:AB_2629281 | WB (1:5000) |
| Antibody | anti- GAPDH (Mouse monoclonal) | Millipore Sigma | Cat# MAB374, RRID:AB_2107445 | WB (1:10000) |
| Antibody | anti- HSC70 (Mouse monoclonal) | Santa Cruz Biotechnology | Cat# sc-24, RRID:AB_627760 | WB (1:5000) |
| Antibody | anti- p62 (Rabbit polyclonal) | Abcam | Cat# ab109012, RRID:AB_2810880 | WB (1:1000) |
| Antibody | anti- cleaved caspase-3 (Rabbit polyclonal) | Cell Signaling Technology | Cat# 9661S, RRID:AB_2341188 | WB (1:1000) IF (1:500) |
| Antibody | anti- pAMPK (Rabbit polyclonal) | Cell Signaling Technology | Cat# 2535S, RRID:AB_331250 | WB (1:1000) |
| Antibody | anti- AMPK (Rabbit polyclonal) | Cell Signaling Technology | Cat# 5831S, RRID:AB_10622186 | WB (1:1000) |
| Antibody | anti- pS6K (Rabbit polyclonal) | Cell Signaling Technology | Cat# 9234S, RRID:AB_2269803 | WB (1:1000) |
| Antibody | anti- S6K (Rabbit polyclonal) | Cell Signaling Technology | Cat# 2708S, RRID:AB_390722 | WB (1:1000) |
| Sequence-based reagent | SLC7A11_F | This paper | RT-PCR primers | AGGGTCACCTTCCAGAAATC |
| Sequence-based reagent | SLC7A11_R | This paper | RT-PCR primers | GAAGATAAATCAGCCCAGCA |
| Sequence-based reagent | GCLM _F | This paper | RT-PCR primers | CATTTACAGCCTTACTGGGAGG |

*Continued on next page*

*Appendix 1—key resources table continued*

| Reagent type (species) or resource | Designation | Source or reference | Identifiers | Additional information |
|---|---|---|---|---|
| Sequence-based reagent | GCLM _R | This paper | RT-PCR primers | ATGCAGTCAAATCTGGTGGCA |
| Sequence-based reagent | FOXO3_F | This paper | RT-PCR primers | CGGACAAACGGCTCACTCT |
| Sequence-based reagent | FOXO3_R | This paper | RT-PCR primers | GGACCCGCATGAATCGACTAT |
| Sequence-based reagent | NANOG _F | This paper | RT-PCR primers | TTTGTGGGCCTGAAGAAAACT |
| Sequence-based reagent | NANOG _R | This paper | RT-PCR primers | AGGGCTGTCCTGAATAAGCAG |
| Sequence-based reagent | OCT4_F | This paper | RT-PCR primers | TTCAGCCAAACGACCATCTG |
| Sequence-based reagent | OCT4_R | This paper | RT-PCR primers | CACGAGGGTTTCTGCTTTGC |
| Sequence-based reagent | SOX2_F | This paper | RT-PCR primers | GCCGAGTGGAAACTTTTGTCG |
| Sequence-based reagent | SOX2_R | This paper | RT-PCR primers | GGCAGCGTGTACTTATCCTTCT |
| Sequence-based reagent | GLUT1_F | This paper | RT-PCR primers | TATCGTCAACACGGCCTTCACT |
| Sequence-based reagent | GLUT1_R | This paper | RT-PCR primers | AACAGCTCCTCGGGTGTCTTAT |
| Sequence-based reagent | HK2_F | This paper | RT-PCR primers | GCCATCCTGCAACACTTAGGG |
| Sequence-based reagent | HK2_R | This paper | RT-PCR primers | GTGAGGATGTAGCTTG TAGAGGGT |
| Sequence-based reagent | GPI_F | This paper | RT-PCR primers | TGTGTTCACCAAGCTCACAC |
| Sequence-based reagent | GPI_R | This paper | RT-PCR primers | GTAGAAGCGTCGTGAGAGGT |
| Sequence-based reagent | ALDOA_F | This paper | RT-PCR primers | AGGCCATGCTTGCACTCAG |
| Sequence-based reagent | ALDOA_R | This paper | RT-PCR primers | AGGGCCCAGGGCTTCAG |
| Sequence-based reagent | ENO1_F | This paper | RT-PCR primers | GACTTGGCTGGCAACTCTG |

*Continued on next page*

*Appendix 1—key resources table continued*

| Reagent type (species) or resource | Designation | Source or reference | Identifiers | Additional information |
|---|---|---|---|---|
| Sequence-based reagent | ENO1_R | This paper | RT-PCR primers | GGTCATCGGGAGACTTGAAG |
| Sequence-based reagent | LDHA_F | This paper | RT-PCR primers | GGTTGGTGCTGTTGGCATGG |
| Sequence-based reagent | LDHA_R | This paper | RT-PCR primers | TGCCCCAGCCGTGATAATGA |
| Sequence-based reagent | MMP1_F | This paper | RT-PCR primers | ATGCTGAAACCCTGAAGGTG |
| Sequence-based reagent | MMP1_R | This paper | RT-PCR primers | GAGCATCCCCTCCAATACCT |
| Sequence-based reagent | MMP2_F | This paper | RT-PCR primers | ACCAGCTGGCCTAGTGATGATG |
| Sequence-based reagent | MMP2_R | This paper | RT-PCR primers | GGCTTCCGCATGGTCTCGATG |
| Sequence-based reagent | MMP9_F | This paper | RT-PCR primers | ACGCACGACGTCTTCCAGTA |
| Sequence-based reagent | MMP9_R | This paper | RT-PCR primers | CCACCTGGTTCAACTCACTCC |
| Sequence-based reagent | qNCLX_1_F | This paper | RT-PCR primers | GCGTGCTGGTTACCACAGT |
| Sequence-based reagent | qNCLX_1_R | This paper | RT-PCR primers | CCACGGAAGAGCATGAGGAA |
| Sequence-based reagent | qNCLX_2_F | This paper | RT-PCR primers | CCGGCAGAAGGCTGAATCTG |
| Sequence-based reagent | qNCLX_2_R | This paper | RT-PCR primers | ACCTTGCGGCAGTCTACCAC |
| Sequence-based reagent | GAPDH_F | This paper | RT-PCR primers | CCCTTCATTGACCTCAACTACA |
| Sequence-based reagent | GAPDH_R | This paper | RT-PCR primers | ATGACAAGCTTCCCGTTCTC |
| Sequence-based reagent | NONO_F | This paper | RT-PCR primers | TCCGAGGAGATACCAGTCGG |
| Sequence-based reagent | NONO_R | This paper | RT-PCR primers | CCTGGGCCTCTCAACTTCGAT |
| Sequence-based reagent | Tubulin_F | This paper | RT-PCR primers | AGTCCAAGCTGGAGTTCTCTAT |

*Continued on next page*

*Appendix 1—key resources table continued*

| Reagent type (species) or resource | Designation | Source or reference | Identifiers | Additional information |
|---|---|---|---|---|
| Sequence-based reagent | Tubulin_R | This paper | RT-PCR primers | CAATCAGAGTGCTCCAGGGT |
| Sequence-based reagent | G6PD_F | This paper | RT-PCR primers | CGAGGCCGTCACCAAGAAC |
| Sequence-based reagent | G6PD_R | This paper | RT-PCR primers | GTAGTGGTCGATGCGGTAGA |
| Sequence-based reagent | PGD_F | This paper | RT-PCR primers | ATGGCCCAAGCTGACATCG |
| Sequence-based reagent | PGD_R | This paper | RT-PCR primers | AAAGCCGTGGTCATTCATGTT |
| Sequence-based reagent | TKT_F | This paper | RT-PCR primers | TCCACACCATGCGCTACAAG |
| Sequence-based reagent | TKT_R | This paper | RT-PCR primers | CAAGTCGGAGCTGATCTTCCT |
| Sequence-based reagent | g1 | This paper | Guide RNA sequences (*Figure 2A* and *Figure 3—figure supplement 1A*) | GCGCAGATTCAGCCTTCTGC |
| Sequence-based reagent | g2 | This paper | Guide RNA sequences (*Figure 3—figure supplement 1B and C*) | GGGATACTCACGTCTACCAC |
| Sequence-based reagent | g3 | This paper | Guide RNA sequences (*Figure 3—figure supplement 1E*) | GTAGACGTGAGTATCCCGGT |
| Sequence-based reagent | g4 | This paper | Guide RNA sequences (*Figure 3—figure supplement 1E*) | ACCCACACCAGCAGTCCGTC |
| Sequence-based reagent | shRNA (shNCLX#2) | This paper | *Figure 3—figure supplement 1I* | GCCTTCTTGCTGTCATGCAAT |
| Sequence-based reagent | shRNA (shNCLX#3) | This paper | *Figure 3—figure supplement 1I* | GCTCCTCTTCTACCTGAACTT |
| Sequence-based reagent | siRNA (siNCLX) | This paper | *Figure 4—figure supplement 1M* | AACGGCCCCUCAACUGUCUT |
| Sequence-based reagent | NCLX_1 | This paper | PCR primers for screening genomic DNA of NCLX KO clones (*Figure 3—figure supplement 1F*) | GCCAGCATTTGTGTCCATTT |
| Sequence-based reagent | NCLX_2 | This paper | PCR primers for screening genomic DNA of NCLX KO clones (*Figure 3—figure supplement 1F*) | AATTCGTCTCGGCCACTTAC |

*Continued on next page*

*Appendix 1—key resources table continued*

| Reagent type (species) or resource | Designation | Source or reference | Identifiers | Additional information |
|---|---|---|---|---|
| Sequence-based reagent | NCLX_3 | This paper | PCR primers for screening genomic DNA of NCLX KO clones (*Figure 3—figure supplement 1F*) | ACTTAGCACATCGCCACCTG |
| Sequence-based reagent | NCLX_4 | This paper | PCR primers for screening genomic DNA of NCLX KO clones (*Figure 3—figure supplement 1F*) | CTGATCTGCACGCTGAATGG |
| Sequence-based reagent | NCLX_5 | This paper | PCR primers for screening genomic DNA of NCLX KO clones (*Figure 3—figure supplement 1F*) | GAGGTACACAGCAGTTCT CCC |
| Sequence-based reagent | NCLX_6 | This paper | PCR primers for screening genomic DNA of NCLX KO clones (*Figure 3—figure supplement 1F*) | CAGCTGGTGCCCTCAAACAC |
| Sequence-based reagent | PX75 NCLX test F | This paper | PCR primers for screening genomic DNA of NCLX KO HCT116 cells | GTTGTTGAGACAGAGTCTTGC TTC |
| Sequence-based reagent | PX76 NCLX test R | This paper | PCR primers for screening genomic DNA of NCLX KO HCT116 cells | TCCAGCGAGACTGTGCAGAA |
| Sequence-based reagent | px77 | This paper | PCR primers for genotyping NCLX -/- mice | TACAGTCTGGCTCGTTCC CT |
| Sequence-based reagent | px78 | This paper | PCR primers for genotyping NCLX $^{-/-}$ mice | CGGTCCCAGACGCCG T |
| Sequence-based reagent | px79 | This paper | PCR primers for genotyping NCLX $^{-/-}$ mice | CGCTGGGGTCCATCT TTG AT |
| Sequence-based reagent | px80 | This paper | PCR primers for genotyping NCLX $^{-/-}$ mice | TGGGTCTCCGGTCCCAGT A |
| Commercial assay or kit | cDNA Reverse Transcription Kit | Applied biosystems | Cat# 4368814 | |
| Commercial assay or kit | Seahorse XFp Mito Fuel Flex Test Kit | Agilent Technologies | Cat# 103270-100 | |
| Commercial assay or kit | Seahorse XFp Glycolysis Stress Test Kit | Agilent Technologies | Cat# 103017-100 | |
| Commercial assay or kit | Seahorse XFp Cell Mito Stress Test Kit | Agilent Technologies | Cat# 103010-100 | |
| Commercial assay or kit | BCA assay kit | Thermo Fisher Scientific | Cat# A53225 | |
| Commercial assay or kit | TMRE | Thermo Fisher Scientific | Cat# T669 | |
| Commercial assay or kit | CyQUANT | Thermo Fisher Scientific | Cat# C35006 | |

*Continued on next page*

*Appendix 1—key resources table continued*

| Reagent type (species) or resource | Designation | Source or reference | Identifiers | Additional information |
|---|---|---|---|---|
| Chemical compound, drug | Antibiotic and Antimycotic | Thermo Fisher Scientific | Cat# 15240062 | |
| Chemical compound, drug | McCoy's 5A | Corning | Cat# 10-050CV | |
| Chemical compound, drug | RPMI-1640 | Corning | Cat# 10-040CV | |
| Chemical compound, drug | Lipofectamine 2000 | Thermo Fisher Scientific | Cat# 11668019 | |
| Chemical compound, drug | TrypLE | Thermo Fisher Scientific | Cat# 12605028 | |
| Chemical compound, drug | $CoCl_2$ | Sigma-Aldrich | Cat# 15862 | |
| Chemical compound, drug | 2-deoxy-D-glucose (2-DG) | Sigma-Aldrich | Cat# D8375 | |
| Chemical compound, drug | 5- Fluorouracil | Sigma-Aldrich | Cat# F6627 | |
| Chemical compound, drug | Glucose | Sigma-Aldrich | Cat# D9434 | |
| Chemical compound, drug | Puromycin | MP Biomedical | Cat# 02100552 | |
| Chemical compound, drug | RIPA buffer | Sigma | Cat# R0278 | |
| Chemical compound, drug | ATP | Sigma | Cat# A9187 | |
| Chemical compound, drug | Dextrose | Fisher Scientific | Cat# D14 | |
| Chemical compound, drug | Tris Base | Fisher Scientific | Cat# BP152-5 | |
| Chemical compound, drug | NaCl | Fisher Scientific | Cat# S671 | |
| Chemical compound, drug | MOPS SDS running buffer | Thermo Fisher Scientific | Cat# NP0001 | |
| Chemical compound, drug | Tris-Glycine transfer buffer | Bio-rad | Cat#161-0734 | |
| Chemical compound, drug | KCl | Fisher Scientific | Cat# P217 | |

*Appendix 1—key resources table continued*

| Reagent type (species) or resource | Designation | Source or reference | Identifiers | Additional information |
|---|---|---|---|---|
| Chemical compound, drug | $MgCl_2$ | Fisher Scientific | Cat# M33 | |
| Chemical compound, drug | $CaCl_2$ | Fisher Scientific | Cat# C614 | |
| Chemical compound, drug | HEPES | Fisher Scientific | Cat# BP310 | |
| Chemical compound, drug | LDS sample buffer | Thermo Fisher Scientific | Cat# NP0007 | |
| Chemical compound, drug | NuPAGE Bis-Tris precast gels | Thermo Fisher Scientific | Cat# NP0321 | |
| Chemical compound, drug | Polyvinylidene difluoride membrane | Li-Core Biosciences | Cat# 88518 | |
| Chemical compound, drug | Odyssey Blocking Buffer (TBS) | Li-Core Biosciences | Cat# 937-50003 | |
| Chemical compound, drug | Dextran sulfate sodium | MP Biomedical | Cat# 0216011080 | |
| Chemical compound, drug | Azoxymethane | Sigma | Cat# A5486 | |
| Chemical compound, drug | DNase I | Thermo Fisher Scientific | Cat# 18068-015 | |
| Chemical compound, drug | TRIzol | Thermo Fisher Scientific | Cat# 15596018 | |
| Chemical compound, drug | Seahorse XF DMEM Medium pH 7.4 | Agilent Technologies | Cat# 103575-100 | |
| Chemical compound, drug | Seahorse XF 100 mM pyruvate solution | Agilent Technologies | Cat# 103578-100 | |
| Chemical compound, drug | Seahorse XF 200 mM glutamine solution | Agilent Technologies | Cat# 103579-100 | |
| Chemical compound, drug | Seahorse XF 1.0 M glucose solution | Agilent Technologies | Cat# 103577-100 | |
| Chemical compound, drug | Zymogram Developing Buffer (10X) | Thermo Fisher Scientific | Cat# LC2671 | |
| Chemical compound, drug | Zymogram Renaturing Buffer (10X) | Thermo Fisher Scientific | Cat# LC2670 | |

*Continued on next page*

*Appendix 1—key resources table continued*

| Reagent type (species) or resource | Designation | Source or reference | Identifiers | Additional information |
|---|---|---|---|---|
| Chemical compound, drug | Tris-Glycine SDS Running Buffer (10X) | Thermo Fisher Scientific | Cat# LC2675 | |
| Chemical compound, drug | Tween 20 | Fisher Scientific | Cat# BP337 | |
| Software, algorithm | Image J | https://imagej.net/ | RRID:SCR_003070 | |
| Other | DAPI | Sigma-Aldrich | Cat# D9542 | 1µg/ml |
| Other | Hoechst | Thermo Fisher Scientific | Cat# H3570 | 1µg/ml |
| Other | IRDye 800CW Goat anti-Mouse | Li-Core Biosciences | Cat# 925-32210 | 1:10000 |
| Other | IRDye 800CW Donkey anti-Rabbit | Li-Core Biosciences | Cat# 925-32213 | 1:5000 |
| Other | MitoSox Red | Thermo Fisher Scientific | Cat# M36008 | |
| Other | Mito TEMPO | Thermo Fisher Scientific | Cat# SML0737 | |
| Other | Mito Tracker Green FM | Thermo Fisher Scientific | Cat# M7514 | |
| Other | Mito Tracker Deep red FM | Cell Signaling Technology | Cat# 8778S | |
| Other | Fura-2 AM | Thermo Fisher Scientific | Cat# F1221 | |
| Other | FluoroBlok | Corning | Cat# 351152 | |
| Other | BioCoat Tumor Invasion Plate | Corning | Cat# 80774380 | |
| Other | SYBER select master mix | Thermo Fisher Scientific | Cat# 4472920 | |
| Other | Novex 10% Zymogram Plus (Gelatin) Protein Gels, 1.0 mm, 10-well | Thermo Fisher Scientific | Cat# ZY00100 | |
| Other | SimplyBlue Safe Stain | Thermo Fisher Scientific | Cat# LC6060 | |

