## [Decision Letter]

**Acceptance summary:**

The authors tell a convincing story about the functional consequences of Na/Ca/Li exchanger (NCLX) downregulation in colorectal cancer cells (CRC), showing that, although loss of NCLX suppresses cancer cell proliferation, it also enhances cell migratory behavior. Thus, the NCLX downregulation in CRC exerts a net detrimental effect owing to an associated increase in metastatic potential. Overall, these findings significantly advance our understanding of the contribution of mitochondrial Ca^2+^ dynamics to CRC progression and suggest NCLX as a potential therapeutic target

**Decision letter after peer review:**

Thank you for submitting your article "Dichotomous role of the mitochondrial Na+/Ca^2+^/Li+ exchanger NCLX in colorectal cancer growth and metastasis" for consideration by *eLife*. Your article has been reviewed by three peer reviewers, including Mark T Nelson as the Reviewing Editor and Reviewer #1, and the evaluation has been overseen by a Reviewing Editor and Richard Aldrich as the Senior Editor.

The reviewers have discussed the reviews with one another and the Reviewing Editor has drafted this decision to help you prepare a revised submission.

Summary:

In their manuscript, Pathak/Gueguinou et al. investigate the role of the mitochondrial Na+/Ca^2+^ exchanger, NCLX, in colorectal cancer (CRC) cell proliferation and invasion. The primary experimental models used to address these questions include colon cancer cell lines (HCT226, DLD1) in which NCLX was constitutively knocked out (CRISPR/Cas9) or transiently knocked down (siRNA), as well as NCLX-/- mice in which NCLX was knocked out globally using the CRISPR/Cas9 system. The authors found that NCLX expression is downregulated in colorectal tumors from CRC patients, exhibiting a greater decrease in advanced-stage patients, and showed that NLCX knockout in mice resulted in fewer, smaller tumors in a DSS-induced colonic tumor model. NLCX knockout/knockdown in vitro caused mitochondrial Ca^2+^ (mtCa^2+^) overload owing to abrogated NLCX-mediated mtCa^2+^ efflux in the context of largely unchanged MCU-mediated Ca^2+^ influx into mitochondria and, as a consequence, led to overproduction of reactive oxygen species (ROS). It also disrupted mitochondrial membranes and cristae, induced expression of mitophagy markers, caused mitochondrial membrane depolarization, and negatively affected respiratory reserve. Notably, they found that NLCX knockout/knockdown suppressed proliferation of CRC cells, but enhanced CRC cell migration; it also increased chemoresistance and expression of genes involved in epithelial-to-mesenchymal transition and hypoxia (i.e., hypoxia-inducible factor 1α [HIF1α]), as well as stem cell-related genes. The authors conclude that loss of NCLX expression and accompanying mitochondrial Ca^2+^ overload drives the metastatic behavior of CRC cells through ROS-dependent activation of HIF1α and altered expression of HIF1α target genes.

Essential revisions:

Overall

Using a broad range of experimental approaches, the authors tell a convincing story about the functional consequences of NCLX downregulation in CRC, showing that, although loss of NCLX suppresses cancer cell proliferation, it also enhances cell migratory behavior. Thus, the NCLX downregulation in CRC exerts a net detrimental effect owing to an associated increase in metastatic potential. The experimental models (wild-type and NCLX-KO/KD CRC cells, wild-type and NCLX-/- mouse DSS-induced colonic tumor models, patient-derived CRC tumors) are complementary and generally appropriate, and the conclusions follow logically from the results. Overall, these findings significantly advance our understanding of the contribution of mitochondrial Ca^2+^ dynamics to CRC progression and suggest NCLX as a potential therapeutic target.

General comments

A unified approach should be applied to the presentation of results from the various HT116 and DLD1 NCLX-KO clones. In some cases, results from all NCLX-KO clones are shown in figure panels, but in others, results from only a subset (or even just one) clone are presented. Results from all clones should be shown throughout or results from a representative clone (or clones) of each cell type should be consistently shown. Choosing to show results from one clone but not others in some experimental settings leaves the impression of cherry picking.

Awkward phrasing and grammatical errors are frequent enough to significantly impact readability and occasionally obscure meaning. Recommend additional editing efforts.

1) Only circumstantial evidence supports the claim that NCLX expression controls tumor progression via matrix calcium. It has to be shown that MCU deletion prevents the effect of NCLX in colorectal tumor cells in at least in one of the paradigms. The Authors propose that the effect of NCLX targeting on the tumor growth and metastasis formation is mediated through mitochondrial matrix [Ca^2+^]. However, an alternative possibility is that NCLX per se has a function in the tumors. Studying the MCU-deficient cells would help to discriminate between these distinct possibilities. Losing the tumor growth/metastasis effects of NCLX targeting in MCU-deficient cells would directly support the Authors' proposal, and the opposite result would be conclusive evidence that NCLX is relevant for the tumors independent of matrix calcium signaling.

After discussion, the reviewers felt that conclusion should be rephrased to leave open the possibility that the NCLX effect is not mediated through matrix [Ca^2+^].

2) Although the results of Figure 2 are very interesting, the reasons that an intrasplenic injection was chosen are not explained. In this model, metastasis to the colon was measured and not the other way around. Some discussion of why this injection site was chosen and of the limitations of this system would be helpful to better appreciate this experiment.

3) In Figure 3—figure supplement 1L, there appears to be a several-fold increase in cleaved caspase 3 positive cells. For some reason, this is reported as a slight increase in apoptosis and then discounted. I don't quite understand the reasons for this conclusion. The average number of cells exhibiting cleaved caspase 3 increases from less than 10% to around 30%. Why can't this be responsible for the decrease in total cell number over time? Particularly when you consider the growth curves were not logarithmic, which could be explained by apoptosis?

4) Also, in Figure 3, there is a statement that differences in MMP activity are greater than differences in MMP expression. This statement can be easily tested statistically; please do so or remove the statement.

5) The basis for the statement that mitophagy is increased in NCLX-KO cells is unclear to me based on the data shown in Figure 4 and Figure 4—figure supplement 2. Evidence is provided that mitochondria are damaged and exhibit less cristae. It is shown that there is more LC3BII, which could indicate the presence of more autophagic vesicles, however, there is also increased P62 expression in Figure 4; P62 is degraded during autophagy. There is also no loss of total mitochondria as shown in Figure 4D. As such, while these data definitely show mitochondrial damage, I do not understand the reasons for attributing it to increased mitophagy.

6) In the Discussion section, there is a statement that reduced expression of cell cycle-related genes in NCLX-KO cells is consistent with decreased proliferation. Does this refer to decreased expression of genes associated with the G2/M checkpoint? If so, the data shown isn't entirely sufficient to make this claim; are the genes that are downregulated within the G2/M checkpoint category all genes that promote proliferation?

7) Figure 3—figure supplement 1L: Was the increase in cleaved caspase-3 detected in all three HT116 clones, or was only clone #33 tested?

8) In the absence of more direct approaches for assessing apoptosis, the suggestion that apoptosis is generally increased in NCLX-KO CRC cells based on an increase in cleaved caspase-3 (as implied by the text) is an overinterpretation.

9) "Collectively, these results show that the loss of NCLX in CRC cells causes an increase in migration and invasion of CRC cells through increased MMP1, 2, and 9 protein levels and activity." This conclusion should be toned down. No experiments capable of establishing a causal relationship between MMP increases and migration/invasion (MMP KO, MMP inhibitors, etc.) were done. At best, these expression and zymography data are suggestive of the involvement of MMPs.

10) Figure 3: Despite the fact that MMP1 showed the strongest increase in protein expression among the three MMPs tested (Figure 3L,M), zymography results are only shown for *MMP2* and MMP9. Why?

11) Loss of NCLX in CRC cells inhibits mtCa^2+^ extrusion, causes mitochondrial perturbations, and enhances mitophagy and mitochondrial ROS.

12) "While basal and ATP-dependent OCR were slightly decreased…"

No. In the main figure (Figure 4K), basal OCR was significantly decreased whereas ATP-dependent OCR was not. "Slightly decreased" is a vague phrase that cannot be used to simultaneously describe both significant and non-significant differences (any "difference" that does not reach statistical significance should be described as a trend or tendency).

13) Although the different DLD1 clones yielded OCR results that were generally consistent with each other (Figure 4—figure supplement 2), exhibiting significant decreases in basal, spare, ATP-dependent and max OCR, they differed from those of clone #33 HCT116 NCLX-KO cells with respect to ATP-dependent OCR (Figure 4K). Curiously, although results for all HCT116 NCLX-KO clones were presented in Figure 4A,D,E,F,G,H and I, data for HCT116 NCLX-KO clones #37 and #59 were not shown in Figure 4K, making it difficult to gauge the significance of this discrepancy. These additional data should be shown.

14) "non-significant reduction" is an oxymoron: if it isn't significant, it isn't a reduction. Modifying this phrase with "mild" just further confuses the issue.

15) In the seahorse experiments, every parameter seems to be lower in the NCLX-KO (basal, oligomyicin, uncoupled and even after Rotenone/Antimycin), this raises the concern that such a reduction could be caused simply by a lower total number of cells at the moment of the experiment (due to attachment, growth, cell death, etc) or lower number of mitochondria. While the latter is trickier to assess, the former can be done by washing the wells in the seahorse plates with PBS to remove dead/detached cells and then doing protein extraction from the wells with RIPA buffer (and use that to normalize the values rather than the amount of cells plated, which could vary between the groups due to cell attachment, death and growth during the short time they stay in culture in the plates).

16) Why is colorectal cancer chosen to study the effect of NCLX expression? Did the authors have some prior knowledge or some supporting data of the involvement of NCLX in colorectal cancer specifically?

Do the risk factors of colorectal cancer like smoking, alcohol consumption or genetic predisposition have any effect on NCLX expression levels? From the data set available to the authors or the patient samples available (if the patient background history is available), can something be concluded in this respect?

17) NCLX loss (1) inhibits proliferation and primary tumor growth, while (2) enhancing metastasis, and drug resistance, suggesting that a loss of NCLX contributes to CRC metastatic progression. How is the expression of NCLX different (if any) in the metastatic part of the tumor (i.e. at the secondary site of the tumor)?

18) Do all samples obtained from patients with colorectal cancer or the information available in the database have NCLX dowregulation? If no, then what percentage of the available dataset has this NCLX downregulation?

---

## [Author Response]

Essential revisions:OverallUsing a broad range of experimental approaches, the authors tell a convincing story about the functional consequences of NCLX downregulation in CRC, showing that, although loss of NCLX suppresses cancer cell proliferation, it also enhances cell migratory behavior. Thus, the NCLX downregulation in CRC exerts a net detrimental effect owing to an associated increase in metastatic potential. The experimental models (wild-type and NCLX-KO/KD CRC cells, wild-type and NCLX-/- mouse DSS-induced colonic tumor models, patient-derived CRC tumors) are complementary and generally appropriate, and the conclusions follow logically from the results. Overall, these findings significantly advance our understanding of the contribution of mitochondrial Ca^2+^ dynamics to CRC progression and suggest NCLX as a potential therapeutic target.General commentsA unified approach should be applied to the presentation of results from the various HT116 and DLD1 NCLX-KO clones. In some cases, results from all NCLX-KO clones are shown in figure panels, but in others, results from only a subset (or even just one) clone are presented. Results from all clones should be shown throughout or results from a representative clone (or clones) of each cell type should be consistently shown. Choosing to show results from one clone but not others in some experimental settings leaves the impression of cherry picking.

To address this concern, we have performed additional studies for the missing HCT116 and DLD1 clones. Now we show results from all 6 clones (3 clones of each HCT116 and DLD1 cells). These are now shown in Figure 3—figure supplement 1L; Figure 4L; Figure 4M; Figure 4—figure supplement 2F and Figure 5—figure supplement 2G.

Awkward phrasing and grammatical errors are frequent enough to significantly impact readability and occasionally obscure meaning. Recommend additional editing efforts.

Thank you. We have taken every effort to correct typos and grammatical errors in the manuscript.

1) Only circumstantial evidence supports the claim that NCLX expression controls tumor progression via matrix calcium. It has to be shown that MCU deletion prevents the effect of NCLX in colorectal tumor cells in at least in one of the paradigms. The Authors propose that the effect of NCLX targeting on the tumor growth and metastasis formation is mediated through mitochondrial matrix [Ca^2+^]. However, an alternative possibility is that NCLX per se has a function in the tumors. Studying the MCU-deficient cells would help to discriminate between these distinct possibilities. Losing the tumor growth/metastasis effects of NCLX targeting in MCU-deficient cells would directly support the Authors' proposal, and the opposite result would be conclusive evidence that NCLX is relevant for the tumors independent of matrix calcium signaling.

We have extensively studied the role of MCU in colorectal cancer progression and found that MCU deficient cells present the opposite phenotype (i.e. reduced metastasis) as compared to NCLX deficient CRC cells. These extensive data on MCU, which validate the role of mtCa^2+^ in CRC, are substantial and warrant their own manuscript. We are in the process of finalizing a manuscript on the role of MCU in colorectal cancer progression.

After discussion, the reviewers felt that conclusion should be rephrased to leave open the possibility that the NCLX effect is not mediated through matrix [Ca^2+^].

A statement has been added in the Discussion section stating that the role of MCU has to be tested in CRC progression to establish that mtCa^2+^ is required CRC growth.

2) Although the results of Figure 2 are very interesting, the reasons that an intrasplenic injection was chosen are not explained. In this model, metastasis to the colon was measured and not the other way around. Some discussion of why this injection site was chosen and of the limitations of this system would be helpful to better appreciate this experiment.

The main aim of the xenograft model is to understand the role of NCLX in regulating metastatic property of colorectal cancer in vivo. No xenograft model is 100% perfect. Intrasplenic injection is a well-established method to study metastasis to the liver for example. The advantage of this model is that we can monitor metastasis to both colon and liver. The only known limitation of this model is that the overall efficiency of liver metastasis and colonization is low (Heijstek, Kranenburg and Borel Rinkes, 2005). There is a model that tests metastasis from the colon to distant sites but this model is not within the expertise of our laboratory. A couple of sentences have been added to the Results section on page 6 to explain why we used this model.

3) In Figure 3—figure supplement 1L, there appears to be a several-fold increase in cleaved caspase 3 positive cells. For some reason, this is reported as a slight increase in apoptosis and then discounted. I don't quite understand the reasons for this conclusion. The average number of cells exhibiting cleaved caspase 3 increases from less than 10% to around 30%. Why can't this be responsible for the decrease in total cell number over time? Particularly when you consider the growth curves were not logarithmic, which could be explained by apoptosis?

We completely agree that the cleaved caspase 3 staining in NCLX KO cell was increased significantly and this is a clear indication of apoptosis that might contribute to the reduced tumor size. We have added the cleaved caspase 3 staining quantification from the other two clones #37 and #59 of HCT116 NCLX KO cells. These clones also show a significant increase in cleaved caspase3 staining. However, we were not able to confirm this result by western or Annexin V staining, because we discovered from our metabolomics screen that the lipid profile, including phosphatidylserine levels were altered in NCLX KO cells. Therefore, we think that apoptosis has a role in reduced tumor phenotype of NCLX KO cells, but it might not be the only player in this particular case.

4) Also, in Figure 3, there is a statement that differences in MMP activity are greater than differences in MMP expression. This statement can be easily tested statistically; please do so or remove the statement.

The statement has been removed.

5) The basis for the statement that mitophagy is increased in NCLX-KO cells is unclear to me based on the data shown in Figure 4 and Figure 4—figure supplement 2. Evidence is provided that mitochondria are damaged and exhibit less cristae. It is shown that there is more LC3BII, which could indicate the presence of more autophagic vesicles, however, there is also increased P62 expression in Figure 4; P62 is degraded during autophagy. There is also no loss of total mitochondria as shown in Figure 4D. As such, while these data definitely show mitochondrial damage, I do not understand the reasons for attributing it to increased mitophagy.

We agree that the increased LC3BII and p62 levels clearly indicate that the there is an increase of autophagosome formation, and this does not mean that there is increased mitophagy in NCLX KO cells. Electron micrograph images shows that NCLX KO cells have significantly more autophagosomes than control HCT116 cells. While the number of mitophagosomes was increased in NCLX KO clones, this increase was not statistically significant compared to control HCT116 cells. Therefore, we have modified the interpretation of Figure 4 and added statements in the Discussion section, indicating that more experiments are required to determine the fate of damaged mitochondria of NCLX KO cells.

6) In the Discussion section, there is a statement that reduced expression of cell cycle-related genes in NCLX-KO cells is consistent with decreased proliferation. Does this refer to decreased expression of genes associated with the G2/M checkpoint? If so, the data shown isn't entirely sufficient to make this claim; are the genes that are downregulated within the G2/M checkpoint category all genes that promote proliferation?

Yes, the genes shown in the heatmap in Figure 5—figure supplement 1D positively regulate cell proliferation or regulates transition from G2/M. Reduction in any of those genes will lead to reduced proliferation or cell cycle arrest. However, we acknowledge that the RNAseq data must be validated by qRT-PCR or western to confirm the downregulation of these genes.

7) Figure 3—figure supplement 1L: Was the increase in cleaved caspase-3 detected in all three HT116 clones, or was only clone #33 tested?

In the first version of the manuscript, only one clone was tested. However, we now have stained the clone #37 and #59 for cleaved caspase-3 and added the data in Figure 3—figure supplement 1L.

8) In the absence of more direct approaches for assessing apoptosis, the suggestion that apoptosis is generally increased in NCLX-KO CRC cells based on an increase in cleaved caspase-3 (as implied by the text) is an overinterpretation.

Yes, we agree with the reviewer’s suggestion and have adjusted our conclusion. We have added statements in the Discussion section, stating that more direct approaches are required to confirm that the deletion of NCLX causes apoptosis in CRC cells.

9) "Collectively, these results show that the loss of NCLX in CRC cells causes an increase in migration and invasion of CRC cells through increased MMP1, 2, and 9 protein levels and activity." This conclusion should be toned down. No experiments capable of establishing a causal relationship between MMP increases and migration/invasion (MMP KO, MMP inhibitors, etc.) were done. At best, these expression and zymography data are suggestive of the involvement of MMPs.

We have toned down the conclusion and changed the word “show that” to “suggest that” in subsection “Loss of NCLX inhibits proliferation but enhances migration and invasion of CRC cells” in results related to Figure 3.

10) Figure 3: Despite the fact that MMP1 showed the strongest increase in protein expression among the three MMPs tested (Figure 3L,M), zymography results are only shown for MMP2 and MMP9. Why?

Both *MMP2* and MMP9 degrade gelatin and their activity can be tested by using publicly available gelatin zymograms. MMP1 does not act on gelatin, but on collagen. Therefore, collagen zymograms are required to test the activity of MMP1. The collagen zymograms are not available for purchase and making theses collagen zymograms is beyond the expertise of our lab.

11) Loss of NCLX in CRC cells inhibits mtCa^2+^ extrusion, causes mitochondrial perturbations, and enhances mitophagy and mitochondrial ROS.

We agree with this comment. We have removed the term mitophagy from the sentence.

12) "While basal and ATP-dependent OCR were slightly decreased…"No. In the main figure (Figure 4K), basal OCR was significantly decreased whereas ATP-dependent OCR was not. "Slightly decreased" is a vague phrase that cannot be used to simultaneously describe both significant and non-significant differences (any "difference" that does not reach statistical significance should be described as a trend or tendency).

We have changed the statement in the Results section.

13) Although the different DLD1 clones yielded OCR results that were generally consistent with each other (Figure 4—figure supplement 2), exhibiting significant decreases in basal, spare, ATP-dependent and max OCR, they differed from those of clone #33 HCT116 NCLX-KO cells with respect to ATP-dependent OCR (Figure 4K). Curiously, although results for all HCT116 NCLX-KO clones were presented in Figure 4A,D,E,F,G,H and I, data for HCT116 NCLX-KO clones #37 and #59 were not shown in Figure 4K, making it difficult to gauge the significance of this discrepancy. These additional data should be shown.

We have added data on HCT116 NCLX KO #37 and #59 in Figure 4. Similar to HCT116 NCLX KO #33, the ATP-dependent OCR remains unaltered in HCT116 NCLX KO #37 and #59.

14) "non-significant reduction" is an oxymoron: if it isn't significant, it isn't a reduction. Modifying this phrase with "mild" just further confuses the issue.

We have changed the sentence from non-significant reduction in proliferation to “it did not affect the proliferation” in subsection “NCLX deficiency causes stem cell-like phenotype and chemoresistance of CRC cells”.

15) In the seahorse experiments, every parameter seems to be lower in the NCLX-KO (basal, oligomyicin, uncoupled and even after Rotenone/Antimycin), this raises the concern that such a reduction could be caused simply by a lower total number of cells at the moment of the experiment (due to attachment, growth, cell death, etc) or lower number of mitochondria. While the latter is trickier to assess, the former can be done by washing the wells in the seahorse plates with PBS to remove dead/detached cells and then doing protein extraction from the wells with RIPA buffer (and use that to normalize the values rather than the amount of cells plated, which could vary between the groups due to cell attachment, death and growth during the short time they stay in culture in the plates).

We agree with the reviewer’s comment. We have added new seahorse experiments from HCT116 NCLX KO #37 and #59 clones in Figure 4L,M. In this experiment we have normalized the OCR values with the protein amount. After normalization, we observe a significant reduction in spare respiratory capacity and maximal respiration in HCT116 NCLX KO #37and #59 clones. However, basal respiration appears to be similar to that of HCT116 control.

16) Why is colorectal cancer chosen to study the effect of NCLX expression? Did the authors have some prior knowledge or some supporting data of the involvement of NCLX in colorectal cancer specifically?Do the risk factors of colorectal cancer like smoking, alcohol consumption or genetic predisposition have any effect on NCLX expression levels? From the data set available to the authors or the patient samples available (if the patient background history is available), can something be concluded in this respect?

When we analyzed the TCGA data of colorectal cancer patient to determine the differential expression of calcium handling proteins, we observed a significant reduction of NCLX mRNA levels in the patient samples as compared to adjacent normal tissue. Unfortunately, we did not observe a correlation between smoking, alcohol consumption, genetic predisposition and NCLX mRNA levels in colorectal cancer patient samples.

17) NCLX loss (1) inhibits proliferation and primary tumor growth, while (2) enhancing metastasis, and drug resistance, suggesting that a loss of NCLX contributes to CRC metastatic progression. How is the expression of NCLX different (if any) in the metastatic part of the tumor (i.e. at the secondary site of the tumor)?

We have analyzed the NCLX mRNA levels in primary tumors from patients both at early and late stages of CRC. We could not analyze NCLX mRNA levels in the secondary site of tumors because tissues from secondary tumors of CRC patients are not readily available.

18) Do all samples obtained from patients with colorectal cancer or the information available in the database have NCLX dowregulation? If no, then what percentage of the available dataset has this NCLX downregulation?

Yes, in the case of patient samples, 100% of tumor tissues analyzed had a significant reduction in NCLX mRNA level compared to their respective adjacent normal tissue. The data are represented in Figure 1B and 1G.